# Characterizing Nonlinear Dynamics via Smooth Prototype Equivalences

## Abstract

Characterizing the long term behavior of dynamical systems given limited measurements is a common challenge throughout the physical and biological sciences. This is a challenging task due to transient variability in systems with equivalent long-term dynamics. We address this by introducing smooth prototype equivalences (SPE), a framework for matching sparse observations to prototypical behaviors using invertible neural networks which model smooth phase space deformations. SPE can localize the invariant sets describing long-term behavior of the observed dynamics through the learned mapping from prototype space to data space. Furthermore, SPE enables classification of dynamical regimes by comparing the data residual of the deformed measurements to prototype dynamics. Our method outperforms existing techniques in the classification of oscillatory systems and can efficiently identify invariant structures like limit cycles and fixed points in an equation-free manner, even when only a small, noisy subset of the phase space is observed. Finally, we show how our method can be used for the detection of biological processes like the cell cycle trajectory from high-dimensional single-cell gene expression data.

## 1 Introduction

Predicting the long-term behavior of dynamical systems from sparse data is a challenging problem in the physical and life sciences (Weisshaar, 2012; Dimitriadis, 2017; Chervov & Zinovyev, 2022). This situation is further exacerbated in realistic cases where equations are unknown and behavior must be estimated from sparse, noisy or high-dimensional data. To this end, substantial work has been devoted to accurately estimating the vector fields underlying observed sequential data in the form of ordinary differential equations (ODEs) (e.g Chen et al. 2018; Brunton et al. 2016; Farrell et al. 2023; Kochkov et al. 2021; Brunton et al. 2017). However, even when there is access to the full governing equations of the underlying ODE, understanding the overall behavior of the dynamical system is notoriously difficult (Smale, 2000). When only sparse, noisy observations from the vector field are available, even just classifying between a limit cycle or a node attractor in two dimensions becomes a difficult problem (Moriel et al., 2023). Detecting these long-term behaviors can be crucial across different fields; for example, in biology, the cell cycle is often modeled as a limit cycle in gene expression space (Riba et al., 2022; Schwabe et al., 2020; Karin et al., 2023), pinpointing the genes participating in and timings of this fundamental process is the subject of substantial work in computational biology (e.g. Mahdessian et al. 2021; Zheng et al. 2022; Kedziora & Stallaert 2024; Zinovyev et al. 2021; Stallaert et al. 2021). In many such scenarios, it can be useful to leverage domain expertise about the overall dynamical behavior expected in the observed data, like the existence of a limit cycle in the cell cycle example above. If such domain expertise is used, then the search space for invariant structures can be greatly simplified, from all possible structures to only a few which are plausible in the system considered.

We propose a method for determining the type and location of invariant sets from vector field data, subject to constraints introduced by domain expertise. Specifically, we assume that there is a true (hidden) dynamical system governed by an ordinary differential equation (ODE) $\frac{dx}{dt} = f(x)$, and we only have access to sparse observations from this ODE. These observations come in the form of snapshots of the dynamics: pairs of positions $x_i$ and noisy estimates of velocities in those positions, $\dot{x}_i = f(x_i) + \epsilon_i$, where $\epsilon_i$ is a noise vector. To extract information about the long-term behavior of

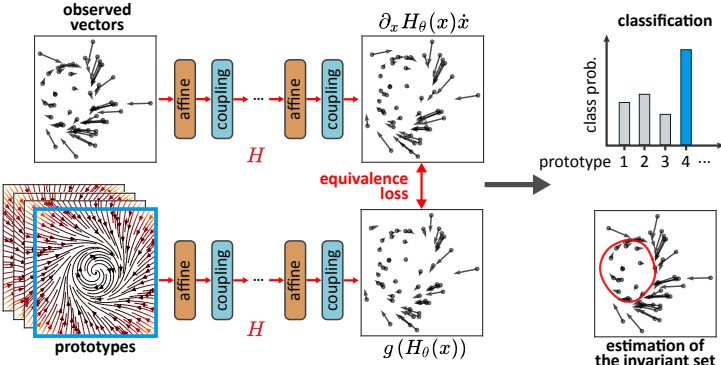

Figure 1: **Schematic for the process of fitting dynamical prototypes using *smooth prototype equivalences* (SPE).** The observed dynamics are compared to each of the chosen prototypes, which involves training an INN $H_\theta(x)$, for each prototype $g(y)$. After training each of the INNs, the observed dynamics are classified as the prototype with the smallest equivalence loss. Given a trained INN $H_\theta(x)$, we have access to a mapping between the data space and the prototype space, which allows us to estimate the long-term behavior, i.e. invariant set, of the dynamics underlying the observed data.

the underlying dynamics, we leverage the fact that the behavior of a dynamical system is not affected by a smooth, invertible change of coordinates. Two systems related by such an invertible mapping are called *smoothly equivalent*, and share the same invariant sets (Chen et al., 2024). The main intuition behind our approach, which we termed *smooth prototype equivalences* (SPE), is to learn an invertible mapping between the data and a *prototypical* version of the behavior which the data is assumed to follow, turning the problem into a form of prototype learning (Rosch, 1973; Bonilla & Robles-Kelly, 2012; Chen et al., 2023).

Our contributions include the following:

- We first demonstrate how SPE can localize limit cycles in a synthetic dataset of 2D dynamical systems drawn from across the physical and life sciences.
- We show how SPE can be used to classify between node and limit cycle attractors, highly abundant and biologically meaningful prototypes, in these synthetic systems, despite the absence of closed-form governing equations. Moreover, we benchmark SPE against previously proposed methods to classify dynamics on a dense grid. Furthermore, SPE, unlike previous methods, is robust to noisy and low-sample conditions, illustrating the compatibility of SPE to real data-driven scenarios.
- We then demonstrate that SPE can naturally extend to more dimensions by applying it to the synthetic six-dimensional repressilator gene regulatory network (Elowitz & Leibler, 2000).
- Finally, we illustrate how SPE can be used in a real scientific scenario, by deploying our method to localize cell-cycle dynamics from noisy, high-dimensional single-cell gene expression measurements, without relying on a gene expression reference atlas.

## 2 RELATED WORKS

**Data driven approaches in dynamical equivalences.** Early notions of equivalence of dynamical systems are due to Poincaré (1892–1899), who introduced foundational ideas like qualitative analysis of trajectories, periodic orbits, and invariant sets. Building on these findings, a large body of work was established to better understand when two systems can be called equivalent (Andronov & Pontryagin, 1937; Hartman, 1960; Sternberg, 1957). Subsequently, in recent years a growing body of work have leveraged relaxed notions of these equivalences (Skufca & Bollt, 2007; 2008) in order to model and compare between nonlinear dynamical systems (Chen et al., 2024; Bramburger et al., 2021; Redman et al., 2022; Glaz, 2024). These characterizations are attractive as they focus on the qualitative behaviors of dynamical systems, such as the stability and type of invariant set. These methodologies

are especially relevant in the field of data-driven scientific discovery, which frequently seeks to compare between experimentally collected dynamical data and analytically derived systems. These approaches, however, typically require explicit access to the functional form of the dynamical systems, a limiting factor in their application to experimentally collected data.

For experimentally collected data, Ostrow et al. introduced DSA which utilizes a modified form of Procrustes analysis in order to align between the delay-embeddings of two observed time series. In this manner, under Taken's embedding theorem (Takens, 1981), the two sequences will only be well aligned when they are conjugate and enough delays are used for the embedding. However, DSA requires sequences of observations long enough for the delay embedding to be beneficial, data which is not always available. Alternatively, Moriel et al. took an empirical approach to the categorization of the invariant set in an observed vector field by training a deep network to distinguish random samples from two conjugacy classes representing oscillatory and non-oscillatory dynamics. This enables classification of systems when the underlying governing equations are hidden, without relying on time series data, but is restricted to conjugacy classes defined during the training of the network and cannot provide more detailed insights beyond the classification of two-dimensional dynamics, such as those related to the structure of the invariant set, or extensions to higher dimensions.

Finally, data-driven approaches to Koopman theory (Brunton et al., 2021) have recently garnered attention, as they enable high-fidelity modeling of high-dimensional dynamics. Such methodologies include extensions of Dynamic Mode Decomposition (Schmid, 2010) from linear to more complex behaviors (Williams et al., 2014; 2015; Li et al., 2017; Bevanda et al., 2022). These methods were utilized with great success to model and analyze complex dynamical systems in many dimensions (Redman et al., 2022) with a strong theoretical basis. However, using these approaches, once the Koopman modes and eigenfunctions are found, more processing must be done in order to further characterize the underlying vector field. For instance finding and categorizing the specific structure of the invariant set remains difficult even after finding the Koopman modes. Moreover, to our knowledge, there is no way to constrain the fitted dynamics to a specific class of dynamical systems.

**Modeling vector field data.** Recent advances in experimental biology and related fields have highlighted the need for improved methods to characterize dynamical systems from noisy data. Increasingly, new techniques generate datasets with partially or completely unordered temporal information. A prominent example is single-cell RNA-sequencing along with RNA velocity (Bergen et al., 2020; La Manno et al., 2018), which defines a vector field in gene expression space and offers insights into cellular dynamics. Another example of note is the changing proportions of different cell types over time, which can be represented as noisy vector field data (Mayer et al., 2023). Often, these systems come with a qualitative understanding of their underlying topology as it often corresponds to classes of physiological or pathological processes (Nitzan & Brenner, 2021; Karin et al., 2023; Zhou et al., 2018; Adler et al., 2020). However, aligning this conceptual model with noisy data remains a challenging task, which is typically pursued separately for each subsystem (Riba et al., 2022; Farrell et al., 2023). Better methods to characterize such data can have enormous potential, but to the best of our knowledge there is currently no unified approach to the analysis of such vector field data.

## 3 METHOD

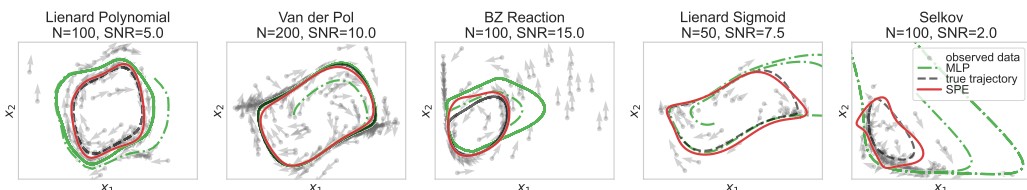

Figure 2: **Estimation of limit cycles using SPE for various 2D dynamical systems.** Examples of invariant sets predicted from observed vectors (in gray), for different dynamical systems which exhibit a limit cycle attractor. For each system, a ground-truth trajectory was simulated from the hidden system and plotted in black. The red curves are the limit cycles predicted using SPE. In all plots, SPE is qualitatively compared to an MLP (in green) trained to reconstruct the vector field.

Throughout this work, we assume that we only observe a sparse subset of the phase space of an ordinary differential equation (ODE), matching observations common to recent advances in computational biology (Bergen et al., 2020; La Manno et al., 2018). Mathematically, let $\mathcal{D} = \{(x_i, \dot{x}_i)\}_{i=1}^{N}$ be a dataset composed of $N$ pairs of positions $x_i \in \mathbb{R}^d$ and their respective velocities $\dot{x}_i \in \mathbb{R}^d$. We assume that these velocities originate from a *hidden* ground-truth ODE and that we only have access to noisy measurements of the true velocities.

### 3.1 SMOOTH PROTOTYPE EQUIVALENCES

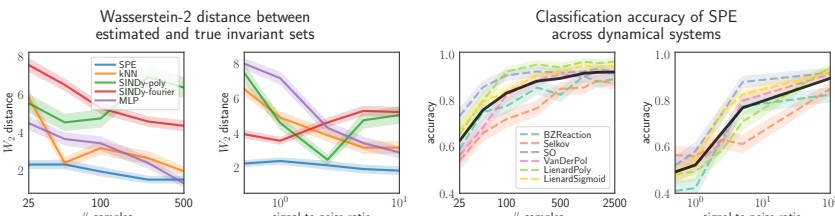

Figure 3: **Wasserstein distance (left) and classification accuracy (right) as a function of the number of observed points and amount of observation noise. Left:** W2 distance between the estimated and true invariant sets, averaged over 1000 randomly sampled datasets (see Appendix C.2), for each method. When changing the number of samples the SNR is held at 2.5, and conversely $N = 100$ points were used when evaluating performance with varying amounts of noise. Shaded areas correspond to standard error of the mean. **Right:** classification performance of SPE, averaged over 1000 randomly sampled datasets (see Appendix C.4). The shaded areas for the different systems represent the standard error of the mean for that system. The black line is the average accuracy over all systems (simple oscillator (SO), BZ reaction, Selḱov, Liénard sigmoid, Liénard polynomial and Van der Pol). When changing the number of samples the data was kept noiseless, and $N = 500$ points were used when evaluating performance with varying amounts of noise.

In many practical modeling scenarios, there is strong prior information as to the possible behaviors of the data. For instance, cyclic processes such as the cell cycle or the circadian rhythm in single-cell RNA sequencing (scRNA-seq) typically correspond to a limit cycle behavior (Riba et al., 2022; Schwabe et al., 2020; Karin et al., 2023), or hierarchical processes such as the differentiation of cells correspond to convergence to fixed points in gene expression space (Lange et al., 2022; Farrell et al., 2023). We can use this information to effectively shrink the search space when modeling the dynamics of the observed data, constraining the model to approximately follow our prior knowledge. In other words, we can constrain our model to only match dynamical systems similar to a candidate set of governing equations, $\dot{y} = g(y)$. In this scenario, the dynamics of $g(y)$ form a *prototype* for the behavior expected to exist in the observed data.

We utilize *smooth equivalences* between dynamical systems in order to constrain the model to the prototype's dynamics (Chen et al., 2024). Specifically, two systems $\dot{x} = f(x)$ and $\dot{y} = g(y)$ are said to be smoothly equivalent if:

$$\exists H \ \text{s.t.} \ \partial_x H(x) \, \dot{x} = g\left(H(x)\right) \tag{1}$$

where $H : \mathcal{X} \rightarrow \mathcal{Y}$ is a diffeomorphism and $\partial_x H(x)$ is the abbreviated notation for the Jacobian of $H$ with respect to $x$. If two systems obey this equivalence relation, each orbit of the system $f(x)$ has a smooth one-to-one mapping to an orbit in $g(y)$, and thus the two systems share the same qualitative behavior, such as the types and stabilities of their invariant sets. In this work, we parametrize the diffeomorphism using an invertible neural network (INN), $H_\theta(x)$, which is optimized in order to approximately satisfy Equation (1). This is achieved by minimizing the following *equivalence loss* (Chen et al., 2024):

$$L_{\mathrm{E}}(H_\theta, g) = \frac{1}{N} \sum_{i=1}^{N} \left\| \frac{\partial_{x_i} H_\theta(x_i) \dot{x}_i}{\|\partial_{x_i} H_\theta(x_i) \dot{x}_i\|} - \frac{g\left(H_\theta(x_i)\right)}{\|g\left(H_\theta(x_i)\right)\|} \right\|^2 \tag{2}$$

When the equivalence loss is 0, the dynamics that generated $\dot{x}$ and $g(y)$ are smoothly equivalent for any point $x \in \mathcal{D}$. Beyond this, it can be shown that when the two systems are equivalent, lower

values of the equivalence loss correspond to more faithful reconstructions of the underlying vector field (Appendix E.1). This further corresponds to a faithful reconstruction of the invariant set of the underlying vector field if it is structurally stable (Guckenheimer & Holmes, 2013) and under mild conditions on the partial derivatives of the INN with respect to the inputs (Appendix E.2).

In practical scenarios, when the data is both sparse and noisy, even when the two systems are not equivalent the loss will not necessarily be equal to zero. Nonetheless, it was shown that the equivalence loss provides a good indication of similarity between dynamical systems when there is access to a large amount of samples (Chen et al., 2024). To improve convergence, we optimize the diffeomorphism using a regularized version of Equation (2), penalizing transformations with a large determinant and those that push the prototype's invariant set far from the data (see A.5 for details).

### 3.2 INVARIANT SET LOCALIZATION AND PROTOTYPE CLASSIFICATION WITH SPE

Our explicit construction using a mapping between the prototype and the observed data allows us to effectively *localize* structures in the observed data, which are in general very difficult to find (Smale, 2000). As in our running example, suppose that our prototype $g(y)$ has a limit cycle, and we can generate points on this limit cycle $\gamma = \{y_1, \cdots, y_M\}$. Because we learned an explicit mapping between the two spaces, then $H_\theta^{-1}(\gamma) = \left\{H_\theta^{-1}(y_1), \cdots, H_\theta^{-1}(y_M)\right\}$ is guaranteed to be a closed loop in data space. Moreover, if the equivalence loss between the observed data and the prototype is low, then $H_\theta^{-1}(\gamma)$ will be a set of points close to the limit cycle of the true hidden system.

Moreover, SPE allows for classification between a set of prototypes, as the equivalence loss directly corresponds to a measure of dissimilarity. So, given a dictionary of possible prototypes $\{g_k(y)\}_{k=1}^K$, we optimize an INN $H_k(x)$ for each, using the equivalence loss from Section 3.1. We then calculate the equivalence loss, $L_E(H_k, g_k)$, between the observed data and each of the prototypes. If a prototype and the observed data are smoothly equivalent, and the NN is adequately optimized, then the equivalence loss for that prototype will be small. This allows us to classify the observed data according to the prototype that has the smallest equivalence loss.

### 3.3 MODELING DIFFEOMORPHISMS WITH INNs

An important aspect of the equivalence loss from Equation (2) is the differentiable and smooth mapping between data and prototype space. In our work, we use INNs adapted from normalizing flows (NFs, Papamakarios et al. 2021) to model our parametric diffeomorphisms. This choice is guided by the need for efficient and explicit computations of both the log-determinants of the Jacobian as well as the Jacobian-vector products (JVPs), $\partial_x H_\theta(x)\dot{x}$, needed to calculate the equivalence loss.

Our INNs utilize a relatively small number of blocks, composed of alternating invertible linear transformations and affine coupling layers (Dinh et al., 2014). Both of these layer types are adapted so that their JVPs can be calculated in closed form. Specifically, instead of using multi-layer perceptrons (MLPs) typically found in affine coupling layers (Papamakarios et al., 2021) we use cosine features, which we call *Fourier feature coupling*. These Fourier feature coupling layers are highly expressive even when the network is narrow and shallow, and their JVPs can be calculated efficiently and in closed form. For details, see Appendix A.

Other options for invertible mappings exist, such as invertible ResNets (Behrmann et al., 2019; Chen et al., 2024) or Neural ODEs (Chen et al., 2018). However, while these options are typically more expressive than INNs adapted from NFs, they come with steep overheads in the calculations of the inverse function, JVPs and their log-determinants. Because of this, we opted to use sufficiently expressive networks comprised of affine coupling layers.

## 4 RESULTS

### 4.1 RECONSTRUCTING ATTRACTORS FROM SPARSE DATA

First, we demonstrate how SPE can be used to locate attractors in sparse data. This is a challenging problem since sparsity and noise make the underlying, unknown dynamical equations strongly undetermined. We demonstrate how SPE circumvents this difficulty on data simulated from a set

Table 1: **Comparison of the classification accuracy of SPE to different models across various families of dynamical systems, when classifying between periodic and node dynamics.** The classification accuracy is averaged over 100 instanced for each family of system (see Appendix B.1 for details). The observed vectors were positioned on a dense $64 \times 64$ grid, with the addition of Gaussian noise with a standard deviation of $\sigma = 0.1$ on the velocities. Where relevant, the errors in accuracy are one standard deviation, calculated over different initializations. The highest scores for each column are shown in bold, with the second highest underlined.

| | SO | Aug. SO | Liénard Poly | Liénard Sigmoid | Van der Pol | BZ Reaction | Selḱov | Attractor estimation |
|---|---|---|---|---|---|---|---|---|
| SPE | 91±1 | 81±1 | **96±1** | **92±1** | **97±3** | **88±2** | **68±1** | ✓ |
| TWA | **98±1** | **93±1** | 86±13 | **92±7** | 83±15 | 82±11 | 65±3 | ✗ |
| Critical Points | 54 | 56 | 70 | 49 | 56 | 84 | 51 | ✗ |
| Phase2vec | 73±8 | 71±4 | 49±6 | 48±0.1 | 49±4 | 50±0.1 | 49±0.1 | ✗ |
| Autoencoder | 95±3 | 87±1 | 63±16 | 88±9 | 81±16 | 82±13 | 49±2 | ✗ |

of standard families of two-dimensional dynamical systems exhibiting limit cycles. The governing equations for these families of systems can be found in Appendix B.1.

To that end, we match between observed systems and a single limit cycle prototype, chosen to flexibly model point and oscillatory dynamics, which is defined in polar coordinates by:

$$\dot{r} = r(a - r^2), \qquad \dot{\theta} = \omega \tag{3}$$

Here, $a$ is an order parameter that controls whether the system has a node attractor (negative values) or a limit cycle (positive values), and $\omega$ determines the orientation of the cycle (clockwise when negative and counter-clockwise when positive). Data were acquired by sampling initial conditions and points along simulated trajectories (details in Appendix B).

After fitting an INN $H_\theta(x)$ between the observed data $\mathcal{D}$ and our limit cycle prototype, we estimated the location of the limit cycle in data space by uniformly sampling a sequence of points, $T_\mathcal{Y} = \{y_j\}_{j=1}^M$, which lie along the limit cycle in prototype space, $\mathcal{Y}$. Each point was then mapped back into data space to define a trajectory along the estimated invariant set of the observed dynamics: $\hat{T}_\mathcal{X} = H_\theta^{-1}(T_\mathcal{Y}) = \{H_\theta^{-1}(y_j)\}_{j=1}^M$. Examples of predicted limit cycles can be seen in Figure 2, where the black orbits are the true limit cycles underlying the observed vectors while the red orbits are the estimated locations of the same limit cycles by SPE, based on the observed vectors. We observed a tight fit between predicted and true limit cycles in all example systems (Figures 2 and 8).

To more quantitatively evaluate SPE, we calculate the Wasserstein ($W_2$) distance between samples from the invariant set of the ground-truth system and those from the invariant set of the fitted model. To do so, we sample a large number of initial conditions from the phase space of the true system. We then simulate long trajectories starting from these initials conditions, taking the final position of these trajectories as an estimate for samples from the true invariant sets. In this manner, we are also able to compare SPE to baseline methods for estimating vector fields from noisy data. The baselines we compare to SPE are (Appendix C.1): a $k$-**nearest neighbor (kNN) interpolator** which estimates the velocity at a position to be the average velocities of the $k$ nearest neighbors to the position; **SINDy** (Brunton et al., 2016), which approximates the vector as a sparse linear regression with a prespecified dictionary of functions; **MLP**, which approximates the underlying ODE using a multilayer perceptron. Note that while these methods serve as useful baselines for the reconstruction quality of the underlying vector field, unlike SPE they do not explicitly model the invariant sets and do not allow for its direct reconstruction.

When the positions $x_i$ are sparse, or the vectors are noisy, SPE more accurately reconstructs the underlying invariant sets than other baselines (Figure 3). In particular, when the phase space is very sparsely populated or noisy, the baselines tend to estimate a vector field with large or none existent invariant sets, an example of which is shown in Figure 2 (right). In contrast, as SPE is constrained to model only stable dynamics, it is able to more closely reconstruct the underlying vector field.

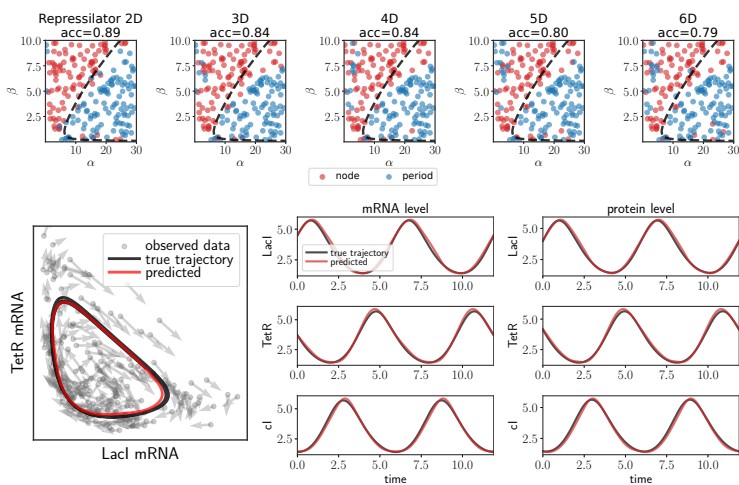

Figure 4: **Classifying dynamics in higher dimensions and recovering the limit cycle from the repressilator system. Top:** each point in the scatter represents a system with specific parameters that was classified by SPE as either a 2D embedded limit cycle (blue) or node attractor (red). The dashed black line depicts the ground truth bifurcation boundary between periodic and node behavior. Each column depicts dynamics projected onto a different dimensionality. **Bottom:** qualitative results of fitting a 6-dimensional repressilator system, projected to the LacI-TetR mRNA plane (left). The gray arrows depict observed position-velocity pairs, $(x_i, \dot{x}_i)$, the black line is the ground-truth invariant set of the dynamics and the red line is the predicted invariant set using SPE. Time series of the gene expression (center) and protein levels (right) can be extracted from the predicted limit cycle (shown in red). These are overlaid on top of a trajectory simulated from the underlying (hidden) system, which was simulated for a long time to ensure convergence to the limit cycle, shown in black.

## 4.2 CLASSIFYING DYNAMICAL SYSTEMS WITH PROTOTYPE EQUIVALENCE

The results of Section 4.1 relied on knowing that the observed system exhibited a particular invariant set. Here, we show that SPE can discover this information directly from the data by fitting multiple prototypes and using the loss to classify the behavior of the dynamics. As our prototype set, we choose all four effective behaviors implied by Equation (3), with $a = \pm\frac{1}{4}$ and $\omega = \pm\frac{1}{2}$. Four diffeomorphisms, $H_k(x)$, were fit to each observed system in parallel and the one with the minimal value of the equivalence loss defined in Equation (2) was used to determine class membership.

We begin by benchmarking SPE next to existing methods for classifying between node or cyclic behaviors on the dataset introduced by Moriel et al. 2023. This dataset is based on simulated families of systems whose order parameters correspond to either a node or a periodic attractor (details for governing equations can be found in Appendix B.1). Accordingly, the task is to classify each system in the dataset as either a point or periodic attractor. Systems in the dataset are comprised of a set of positions $x_i \in \mathbb{R}^2$ and the velocities corresponding to those positions, $\dot{x}_i \in \mathbb{R}^2$, organized on a dense, regular $64 \times 64$ grid for a total of $N = 4096$ observations. Table 1 shows the average accuracy of our method for each family of systems, compared to the following baselines (more details in Appendix C.3): **TWA** (Moriel et al., 2023), a convolutional network trained to classify between periodic and point dynamical systems; **Critical Points** (Helman & Hesselink, 1989; 1991), a heuristic approach that identifies critical points in the vector field; **Phase2vec** (Ricci et al., 2023), a deep learning approach for dynamical systems embedding, which learns vector field representations; and **Autoencoder**, an autoencoder trained to reconstruct the vectors, whose latent code is used to classify the dynamics. SPE outperforms existing baselines across all families of systems except for the family of simple oscillators (SO) and Augmented SO, where TWA, which was trained on thousands of these systems using a dense grid of points in phase space, is more performant (Table 1).

Note that, unlike most of the available baselines, SPE is not restricted to systems defined on a regular grid and can directly localize the invariant set (as shown in Section 4.1). To test the capabilities of SPE in sparse and noisy scenarios, in Figure 3 (right) we track the classification accuracy when decreasing

the number of observed vectors $N$ and under increasingly noisy scenarios, with low signal-to-noise ratios (SNRs) (see Appendix C.4 for details regarding data). The decline in performance is gradual along both of these axes - even when classifying using only 50 random vectors, average accuracy was still relatively high, at $\sim 70\%$.

SPE's classification and localization performance is not limited to limit cycles and node attractors, and can be flexibly generalized to additional prototypes. To demonstrate such capabilities, we evaluated the performance of SPE on a challenging classification of multistable systems with a varying number of basins of attraction (Appendix F.1). Our results show that SPE can successfully distinguish between dynamical systems with varying numbers of fixed points, and can additionally localize them in space (Figure 11). Furthermore, as the loss used to train SPE corresponds to a graded notion of equivalence, we found that SPE is also able to capture dominant behaviors of the system even when there is a mismatch between the true behavior and the chosen prototype. We tested this on the classification of the orientation of the flow in Appendix F.2.

### 4.3 BEYOND TWO DIMENSIONS

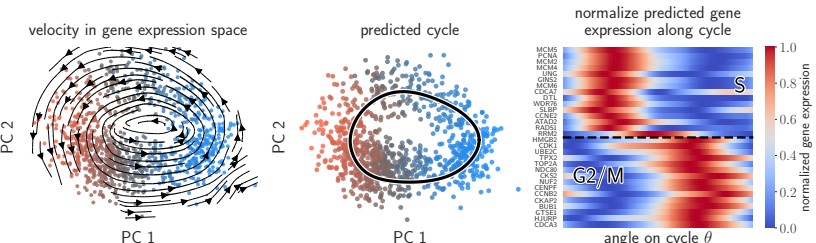

Figure 5: **Recovering the cell cycle gene expression pattern from single-cell data. Left:** gene expression velocity of cells undergoing proliferation in PCA space. Colors denote the cycle phase score based on marker gene expression; red corresponds to high expression of S-phase markers, blue to high expression of G2/M-phase genes, and gray to low levels in both. **Middle:** SPE was used to fit a periodic prototype to the RNA velocity data; the resulting attractor is plotted in black. The cells' locations and colors are as in the plot on the left. **Right:** predicted normalized gene expressions along the cycle attractor predicted by SPE. Specifically, the normalized expression is plotted vs. the angle on the limit cycle ($\theta$), for a set of common marker genes of the S (top of the heatmap) and G2/M (bottom of the heatmap) phases. See details in Appendix D.

Empirical measurements of dynamical systems, whose intrinsic behavior often lies on a low-dimensional manifold, are often embedded in high-dimensional spaces (Champion et al., 2019). Characterizing higher-dimensional systems, whether by localization of invariant structures or by classification of qualitative behaviors from a set of possibilities, is a challenging problem as the location of the lower-dimensional invariant set needs to be identified, and then matched, within the higher dimensional ambient space.

To test our approach in higher-dimensional systems, we first classify and localize invariant structures in data from the repressilator model of gene regulation (Elowitz & Leibler, 2000) (see Appendix B.3 for governing equations). The repressilator models a genetic regulatory network in bacteria, based on the mRNA and protein counts of three genes - TetR, LacI and $\lambda$ phage cI - which inhibit each other in a cyclic fasion. Even though this family of systems spans six dimensions, under specific parameter values it exhibits either a 2D point attractor or contains an embedded limit cycle (Verdugo, 2018a; Potvin-Trottier et al., 2016). As such, to model data from this system we use higher-dimensional prototypes that are the combination of two behaviors: (1) a 2D limit cycle/equilibrium behavior as in Equation (3); (2) exponential decay to a fixed point along variables uncoupled from the embedded 2D system. Mathematically, we define these prototypes as:

$$g(y) = [g_{2D}(y_1), \quad -\tau \cdot y_2]^T \tag{4}$$

with $y_1 \in \mathbb{R}^2$, $y_2 \in \mathbb{R}^{d-2}$, where $g_{2D}(\cdot)$ are velocities in two dimensions, and $\tau$ is a decay factor. We used $\tau = 1/2$ in all of our experiments.

We then classify repressilator systems into node or cycle attractors using SPE, when the systems vary in terms of transcription $\alpha$, and protein/mRNA degradation rates $\beta$. For each system, we simulate $N = 1000$ observed pairs of positions and vectors using the same sampling scheme for the data as described in Section 4.1, Figure 4. To compare the performance along different dimensionalities, we project the observed positions $x_i$ and velocities $\dot{x}_i$ onto a predefined subset of the six dimensions (see Appendix B.3 for details). We observed (Figure 4) that our model can classify two-dimensional repressilator systems better than earlier work (Moriel et al., 2023), whose accuracy was 87% (next to our 89%), and that SPE can accurately distinguish point from cyclic behavior across all dimensionalities considered. Classification of this kind, carried out explicitly in higher dimensionalities, is infeasible using techniques from previous work. Moreover, our optimized diffeomorphisms in the six-dimensional case can be used to accurately locate the cyclic dynamics of the repressilator system, recapitulating the behavior of the system and its invariant set (Figure 4).

### 4.4 UNCOVERING THE CELL CYCLE EXPRESSION PATTERN IN SINGLE-CELL DATA

Finally, we show how SPE can be used to trace the periodic gene expression dynamics of proliferating cells. The expression of cell-cycle genes follows a well-defined trajectory through different phases: initial growth (G1), DNA replication (S phase), further growth (G2), and mitosis (M). Here, we analyze single-cell RNA-sequencing data from a human cell line, profiling over 50,000 genes across more than 1,000 cells (Mahdessian et al., 2021). Previous approaches have inferred cell-cycle progression using marker genes (Tirosh et al., 2016), a learned projection (Zheng et al., 2022; Riba et al., 2022), or using structural priors (Schwabe et al., 2020; Karin et al., 2023), but these do not directly leverage the fact that the cycle has a characteristic dynamical structure in the form of a limit cycle. Here, we show that cell cycle structure can be estimated from RNA velocity (Bergen et al., 2021; Gorin et al., 2022), effective local velocities of cells in gene expression space that are based on differences in spliced and unspliced RNA counts.

Using scVelo (Bergen et al., 2020), we infer high-dimensional RNA velocities for all sampled cells and project them together with their associated gene expression profiles, into a 100-dimensional space using principal component analysis (PCA). The resulting streamlines (Figure 5, left) reveal a noisy trajectory traversing across distinct S-phase and G2/M-phase gene expression states. We then fit a high-dimensional limit-cycle attractor (as described in Equation (4)) to the RNA velocity data, capturing the periodic nature of the cell cycle (Figure 5, middle, see Appendix D for more details on data and prototype). Utilizing the PCA projection, we can map the learned attractor back to gene expression space, revealing periodic, anti-correlated expression patterns of key cell-cycle marker genes for the S and G2/M-phase (Figure 5, right). This demonstrates our method's ability to recover biologically meaningful periodic processes directly from high-dimensional, noisy and sparse experimental data. These results generalize across datasets; in Figure 10 in the appendix, we analyze an additional dataset of proliferating human fibroblasts (Riba et al., 2022), and show that there too SPE is able to localize the cell-cycle from single-cell gene expression and RNA velocity alone, using the same preprocessing and fitting procedure as described for Figure 5 (Appendix D).

## 5 DISCUSSION AND FUTURE DIRECTIONS

Our proposed approach, *smooth prototype equivalences*, addresses a challenging problem in data-driven dynamical systems: characterizing long-term behavior from sparse, noisy data. The key idea is to learn the correspondence between idealized prototype dynamics and their noisy, real-world counterparts in a way that reveals long-term structure. We demonstrated the efficacy and efficiency of this approach on both challenging synthetic and real-world data sets. Furthermore, SPE can be used as a way to classify the long-term dynamics in situations where there are multiple possible ground-truth behaviors.

In this work, our focus was the detection and classification of long-term behaviors that are robust to noise, such as limit cycles and node attractors. Indeed, our main focus were phenomena from empirical biological systems, where dynamics need to be repeatable for proper function. Examples for such dynamics are differentiation of cells into different types (Bergen et al., 2020), the oscillatory process of the circadian clock (Karin et al., 2023), attractors of artificial and biological recurrent neural networks, or the cell cycle (Riba et al., 2022; Mahdessian et al., 2021; Schwabe et al., 2020)

which we explored in Section 4.4, among others. For such systems, which are structurally stable, SPE converges at a faithful reconstruction of the true invariant structures (as shown in Appendix E).

At the other side of this scale are systems which are structurally unstable and, as a special case of particular interest, chaotic systems with strange attractors. SPE was not designed for such scenarios, where even very small perturbations of the underlying vector field can break the structure of the invariant sets they contain. Applying prototype equivalence approaches to chaotic systems, in particular, is an important future challenge, since strange attractors of different fractal dimensions are not smoothly equivalent to each other leaving open the question of how to learn appropriate invariances, especially when subject to experimental noise.

While we have shown that SPE is a useful tool for characterizing of dynamical systems, it explicitly depends on the definition of prototypes which are to be matched to empirical data. SPE makes up for its reliance on pre-defined prototypes, however, with its relaxed notion of equivalence, where a best possible transformation between prototype and data can be found, so that even a partially matching prototype can help localize and categorize dynamical behaviors of scientific interest (demonstrated in Appendix F.2). Empirically, we observe that the equivalence loss can give an ordering on the amount of mismatch between the true underlying dynamics and the chosen prototypes (as shown Appendix F.4).

We expect that in the future SPE can be combined with other methodologies from the field of nonlinear dynamics. For instance, we view our method as complementary to Koopman theory (Brunton et al., 2021), which might allow more diverse dynamics in the operator space. Another extension of interest would be the combination of SPE together with time-delay embeddings, techniques which can be used together in order to elucidate invariant structures in time series data using SPE. Finally, while in this work our primary focus were locally simple behaviors, such as node or cycle attractors or a small collection of node attractors (Appendix F.1), an important future step would entail the combination of multiple local dynamics into a larger, more complex system. One way to do so is the incorporation of SPE in models which explicitly break down space into separate regions, such as mixtures of experts (Jacobs et al., 1991). In such models, SPE can be used as an interpretable component of a more complicated model, capturing intricate dynamics in many dimensions. We also envisage that SPE can be paired with more advanced parameterizations of INNs in order to tackle more complex behaviors, in higher dimensional settings, which may take inspiration from computer vision (Papamakarios et al., 2021; Dinh et al., 2014; Durkan et al., 2019b).

Our method directly addresses challenges faced in scientific domains such as biology, neuroscience, and physics. By offering an equation-free approach for the detection and classification of invariant sets, our method has the potential to advance the understanding of complex, nonlinear processes in systems where analytical modeling is impractical.

## REPRODUCIBILITY STATEMENT

All details necessary to reproduce the results of this work can be found in the appendix:

- Full specifications of our invertible NNs are in Appendix A.
- Hyperparameter choices for all experiments can be found in Appendix A.7.
- Governing equations of all simulated systems are described in Appendix B.1 for 2D systems and Appendix B.3 for the 6D repressilator system.
- Detailed instructions to generate observations from the simulated systems are given in Appendix B.2.

Furthermore, the code will be made publicly available.

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

## A  Normalizing flow details

The diffeomorphisms $H : \mathcal{X} \to \mathcal{Y}$ used in this work are invertible neural networks (INNs) with alternating *Affine* and *Fourier Feature Coupling* (FFCoupling) transforms, both of which are explained

in further details below. Furthermore, the first layer of the normalizing flow was set to standardize the data, effectively an *ActNorm* layer Papamakarios et al. (2021), defined as:

$$[\text{ActNorm(x)}]_i = \frac{x_i - \text{mean}(x_i)}{\text{std}(x_i)} \tag{5}$$

In the above, $\text{mean}(x_i)$ and $\text{std}(x_i)$ are the mean and standard deviations of $i$-th coordinate of the observed data, respectively. This ActNorm layer was kept frozen after initialization, and means that the normalization of the data is explicitly part of the learned diffeomorphism.

## A.1 FOURIER FEATURE COUPLING

The FFCoupling layer we used is a modification of the standard *AffineCoupling* Papamakarios et al. (2021); Dinh et al. (2014) layer used in normalizing flows. AffineCoupling splits an input $x$ to two sub-vectors $x_1$ and $x_2$, and the transformation itself is defined as:

$$\begin{bmatrix} y_1 \\ y_2 \end{bmatrix} = \text{AffineCoupling}(x) = \begin{bmatrix} x_1 \\ \exp[f_s(x_1)] \circ x_2 + f_t(x_1) \end{bmatrix} \tag{6}$$

where $f_s(\cdot)$ is a scaling function, $\exp[f_s(\cdot)]$ is the element-wise exponent of $f_s(\cdot)$, $f_t(\cdot)$ is a translation function, and $\circ$ stands for element-wise multiplication. AffineCoupling is defined in this manner to ensure invertibility, in which case $f_s(\cdot)$ and $f_t(\cdot)$ can be arbitrarily complex functions. These two functions are typically parameterized by a multi-layer perceptron (MLP). Baked into the AffineCoupling transformation is the simple form of its log-determinant:

$$\log|\text{AffineCoupling}(x)| = \sum_i [f_s(x_1)]_i \tag{7}$$

Finally, note that AffineCoupling only transforms $x_2$ as a function of $x_1$. To account for this, it is standard to chain two transformations, one where $x_1$ acts on $x_2$ and another with the roles reversed. For simplicity, we regard such a chaining of two AffineCoupling transformations as a single layer.

Unfortunately, when both the scale and the translation functions are defined as MLPs, calculating the Jacobian or Jacobian-vector products (JVPs) is not straightforward. These JVPs are needed for our loss (Equation (2)) and can potentially be found using autograd methods, albeit at a high computational cost. Furthermore, we expect these diffeomorphisms to typically be quite smooth and act on a range of frequencies, properties which are not easily enforced in MLPs. Instead, we opt to use transformations whose JVPs can be calculated in closed-form and whose properties are well understood.

For our scale and translation functions we use the following *FFCoupling*:

$$\mathcal{F}(x) = \sum_{k=1}^{K} \Theta_k \cos\left(\frac{2\pi}{R} k \cdot x + \phi_k\right) \tag{8}$$

where $\cos(x)$ is an element-wise cosine. As defined, the frequency coefficients $\Theta_k \in \mathbb{R}^{d \times d}$ and phases $\phi_k \in \mathbb{R}^d$ are learnable parameters, while the number of frequencies $K$ and the range $R$ are hyperparameters. When $K$ is large, this transformation is expressive enough to fit any function defined in the range $[-R/2, R/2]$. It also admits a very simple form for the Jacobian, given by:

$$\frac{\partial}{\partial x}\mathcal{F}(x) = -\frac{2\pi}{R} \sum_k k \cdot \Theta_k \text{ diag}\left[\sin\left(\frac{2\pi}{R} k \cdot x + \phi_k\right)\right] \tag{9}$$

## A.2 AFFINE TRANSFORMATION LAYER

Besides the FFCoupling layer, we also use affine transformations. To ensure invertibility, we parameterize this transformation as:

$$\text{Affine}(x) = WW^T x + e^{\varphi} \cdot x + \mu \tag{10}$$

where $W \in \mathbb{R}^{d \times q}$ with some chosen $q$, $\varphi \in \mathbb{R}$ and $\mu \in \mathbb{R}^d$ are learnable parameters. This factorization ensures that the matrix $WW^T + Ie^{\varphi}$ is positive-definite (PD), and thus invertible. It also enables closed-form inversion using the matrix-inversion lemma.

### A.3 HOUSEHOLDER TRANSFORMATIONS

In general, we found that the Affine and FFCoupling layers were expressive enough, when the observed data was mostly aligned with the axes of the prototype. However, in higher dimensions there is no reason to a-priori believe that this should be true. To overcome this difficulty, we used *Householder transformations* (Papamakarios et al., 2021; Tomczak & Welling, 2016), which act as building blocks for orthogonal transformations. The Householder transformation is defined as:

$$\text{HH}(x) = \left( I - \frac{2}{\|v\|^2} v v^T \right) x \tag{11}$$

where $v \in \mathbb{R}^d$ are the parameters of the transformation. This transformation is orthogonal, and so has a determinant of 1. Furthermore, it can be shown (Tomczak & Welling, 2016) that any rotation in a $d$-subspace can be carried out with $d$ of these transformations. Thus, these transforms are useful for more directly rotating space, which is difficult to achieve directly with the Affine transformations defined above.

### A.4 NETWORK AND TRAINING SPECIFICATIONS

As mentioned above, the INNs we use have alternating Affine and FFCoupling transforms. A single block of our network is defined as Affine $\to$ FFCoupling $\to$ ReverseFFCoupling, where the ReverseFFCoupling switches the roles of $x_1$ and $x_2$, as explained in Appendix A.1.

Before these blocks, our networks include a frozen ActNorm layer that standardizes the data and ensures that all learned diffeomorphisms have similar inputs, and then a full-rank Affine transformation, with learnable weights. In the cell-cycle (Section 4.4) and multi-stable dynamics (Appendix F.1) experiments we also added $d$ Householder transforms, where $d$ is the dimension of the data.

### A.5 OPTIMIZATION CRITERIA

In addition to the equivalence loss, $L_E$, we introduced a projection regularizer, $L_{proj}$, that aligns the invariant structures of the prototype with the observed data points. This regularization significantly improves convergence in complex scenarios and promotes identifiability of the solution by encouraging the observed data to lie close to the invariant set implied by the prototype dynamics and the learned diffeomorphism. This regularization is described in full detail in Appendix A.6. Moreover, we found it helpful to regularize the log-determinant to be close to 0, which penalizes networks excessively expanding or contracting space. The full loss used to optimize the diffeomorphism $H$ in SPE is thus given by:

$$L_{tot}(H_\theta, g) = L_E(H_\theta, g) + L_{proj}(H_\theta, g, \lambda_{proj}) + \lambda_{det} L_{det}(H_\theta). \tag{12}$$

All training is carried out using the Adam optimizer Kingma (2014) with weight decay.

### A.6 PROJECTION REGULARIZATION

When using our high-dimensional prototypes from Equation (4), we found that performance was improved by regularizing the diffeomorphism to map the data as close as possible to the invariant sets of the embedded 2D dynamics. Our definition of the high-dimensional prototypes essentially assumes that most of the observed dynamics are at, or close to, a 2D subset of the full space which contains the invariant set of the dynamics. Moreover, our explicit construction of the prototypes, with simple governing equations, allows us to project points in 2D onto their invariant sets. For dynamics of this sort, we expect the positions $x_i$ to be close to the invariant set embedded in a higher dimension. Utilizing this knowledge during optimization can aid convergence, and can be done using our definition of the prototype. Specifically, using the diffeomorphism we can project the points onto the 2D invariant set of the prototype, and then back into data space as follows:

$$P(x; H_\theta) = H_\theta^{-1} \left( [H_\theta(x)]_1, [H_\theta(x)]_2, 0, \cdots, 0 \right) \tag{13}$$

where $[H_\theta(x)]_i$ is the $i$-th coordinate of $H_\theta(x)$ and $P(x; H_\theta)$ denotes the projection according to $H_\theta(x)$.

We model the projection into data space through a Gaussian observation model, with the following loss added when training:

$$L_{\text{proj}}(H_\theta, g, \lambda_{\text{proj}}) = e^{\lambda_{\text{proj}}} \frac{1}{N} \sum_i^N \|x_i - P(x_i; H_\theta)\|^2 - \frac{1}{N}\lambda_{\text{proj}} \tag{14}$$

where $\lambda_{\text{proj}}$ is the regularization coefficient, which takes the form of the log of the precision of the Gaussian observation model. Adding the regularization coefficient in this manner means that it can be optimized in parallel to the diffeomorphism, and allows for an adaptive fitting process. When modeled in this way, the projection is similar to generative topographic mapping (GTM, Bishop et al. (1998)) when the matching between the latent codes and the data is known.

### A.7 HYPERPARAMETERS SETTINGS

For all simulated systems, hyperparameters were chosen to maximize performance on a separate, held out validation set of 100 datasets. During evaluation time, the performance of the methods was not tested on these held out datasets.

**Comparison of classification on dense grids:** for the comparisons in Table 1, we found that weight decay and the other regularizations were not helpful, so all regularization coefficients were set to 0. Additionally, because this data is specified on a dense grid, with the invariant set typically in the center of this grid, we found that the initial ActNorm hurt performance. The initial ActNorm layer in the INN was designed to center data that is assumed to be near the invariant set, but the grid data from Moriel et al. does not follow this assumption. Thus, for this data we did not use the initial ActNorm layer. All networks for those comparisons had 3 blocks with a width of 5 (i.e. using $K = 5$ frequencies) and where trained for 600 iterations with a learning rate of $1e^{-3}$. In all of these, the prototypes were the oscillator prototype defined in Equation (3), with $a = \pm 0.25$ and $\omega = \pm 0.5$.

**Other results on 2D simulated datasets:** for figures Figure 2 and Figure 3, all networks had 4 blocks with a width of 5, and were trained for 500 iterations. For these, we used a learning rate and weight decay of $1e^{-3}$, and the determinant regularization coefficient was $1e^{-3}$. As before, the prototypes used for these were the oscillator prototype with $a = \pm 0.25$ and $\omega = \pm 0.5$.

**Repressilator system:** for Figure 4, we used NFs with 3 blocks and a width of 5. These were trained for 500 iterations, with the same regularization settings as above. Additionally, to fit the systems in more than 2D, we added the projection regularization from Appendix A.6, with the initial regularization $\lambda_{\text{proj}} = e^{-1}$. These were trained for 2000 iterations, and were matched to the high-dimensional versions of the oscillator prototypes from Equation (4), with $a = \pm 0.25$ and $\omega = \pm 2$.

For the hyperparameters used in the cell-cycle experiments, see Appendix D.

As our INNs are quite small, and most of the data is of a low dimension (relative to typical machine learning tasks), a single CPU with 16GB RAM was sufficient for training our networks. Fitting a single INN takes less than 10 seconds for all of the 2D simulated data and around 90-120 seconds for the cell-cycle data.

### A.8 EFFECTS OF HYPERPARAMETERS ON PERFORMANCE

We found that the classification results by SPE are robust to the choice of number of frequencies and layers, as demonstrated in the table below.

In our experiments, we observed that beyond the robust range of SPE demonstrated in Table 2, when many frequencies are used, the networks tend to overfit to overly complex estimations of the invariant set, which are typically not faithful to the true invariant set. These effects can be seen in Figure 6. However, when using too few frequencies and layers, the network is not expressive enough to capture non-elliptical invariant sets. For our experiments in the main text, we chose 4 layers and 5 frequencies as a good balance between classification performance, quality of invariant set localization, and computational efficiency.

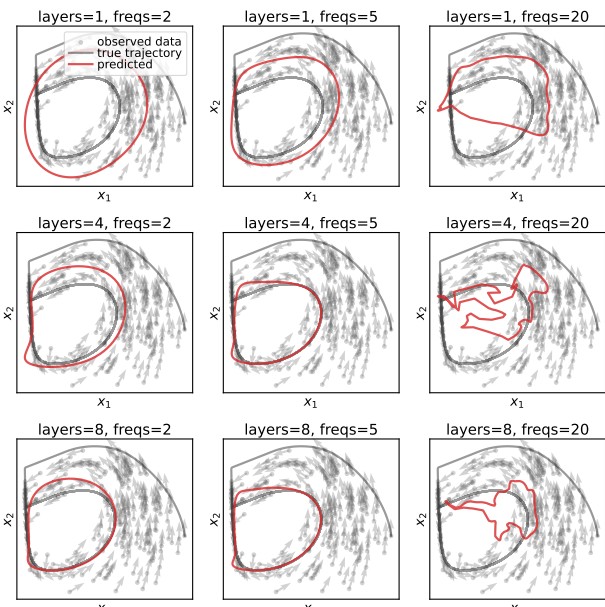

Figure 6: **Estimation of limit cycles using different numbers of layers and frequencies.** Examples of invariant sets predicted (in red) from observed vectors (in gray), for different dynamical systems which exhibit a limit cycle attractor. For each system, a ground-truth trajectory was simulated from the hidden system and plotted in black. The red curves are the limit cycles predicted using SPE.

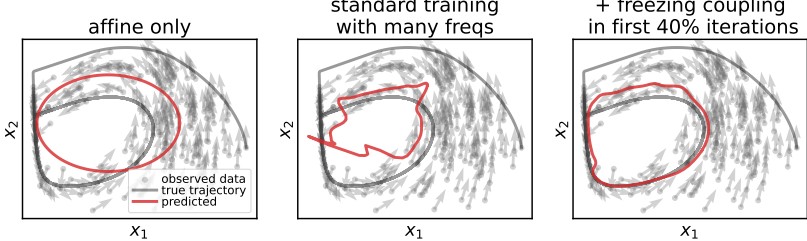

Figure 7: **Initializing network from the solution of an affine-only diffeomorphism improves fitting. Left**: when the invariant set is not an ellipse, using a diffeomorphism defined only by an affine transformation does not faithfully capture the underlying invariant set. **Middle:** if too many frequencies are used with SPE, the model overfits. **Right:** when initializing the model with the result from an affine transformation, SPE converges more faithfully to the true invariant set, even when using many frequencies in the transformation.

Table 2: **Classification performance for different numbers of layers and frequencies of the trained invertible neural network.** Numbers correspond to the average accuracy over 1000 different systems, uniformly sampled from the set of systems from Table 3, with $N = 250$ observations per system.

| frequencies | 2 | 3 | 4 | 5 | 6 | 7 |
|---|---|---|---|---|---|---|
| 2 layers | 88.7% | 84.9% | 89.1% | 85.8% | 84.7% | 81.3% |
| 3 layers | 84.9% | 86.1% | 84.9% | 83.6% | 83.2% | 80.1% |
| 4 layers | 84% | 86.8% | 83.5% | 81.5% | 79.7% | 80.9% |
| 5 layers | 82.2% | 84.9% | 82.5% | 80.3% | 77.8% | 77.3% |

If many frequencies are needed, we found it beneficial to begin training with frozen coupling layers, as shown in Figure 7. This essentially initializes the network at the solution found when the diffeomorphism is constrained to be affine, after which the behavior is fine-tuned using the available frequencies.

# B   SIMULATION DETAILS

## B.1   2D SYSTEMS

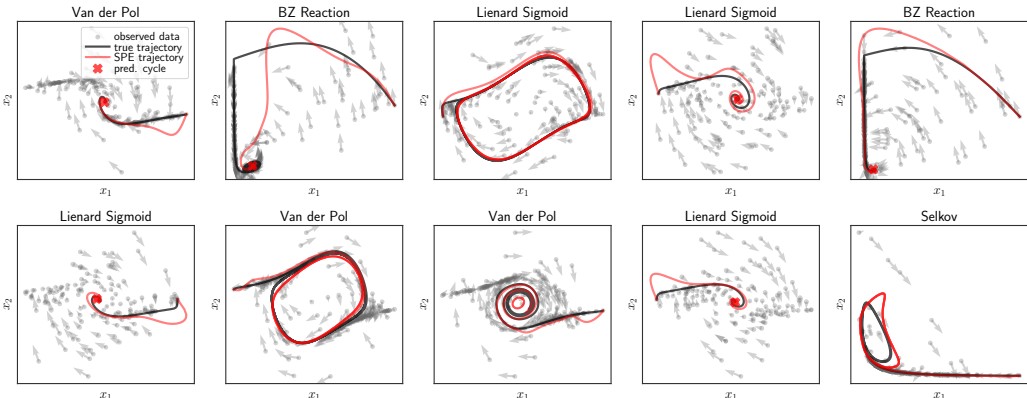

Figure 8: **Randomly picked examples of SPE on 2D synthetic data.** In all of these, notice how SPE is able to localize the invariant set from the data, even when there might be uncertainty regarding the intermediate trajectories.

For the 2D simulated dynamics, we use the same systems as in Moriel et al.. The governing equations for these systems are shown in Table 3 (adapted from Moriel et al. 2023). We additionally study the

Table 3: Governing equations, range of phase space used for initial conditions, range of parameters for simulated systems and parameter settings that are known to correspond with a limit cycle. Adapted from Moriel et al. 2023.

| System Name | Equation | | Initial cond. | Parameter ranges | Cycle condition |
|---|---|---|---|---|---|
| Simple Oscillator (SO) | $\dot{r} = r(a - r^2)$ | $\dot{\theta} = \omega$ | $x_1, x_2 \in [-1, 1]$ | $a \in [-0.5, 0.5], \omega \in [-1, 1]$ | $a > 0$ |
| Liénard Polynomial | $\dot{x}_1 = x_2$ | $\dot{x}_2 = -(ax_1 + x_1^3) - (c + x_1^2)x_2$ | $x_1, x_2 \in [-4.2, 4.2]$ | $a \in [0, 1], c \in [-1, 1]$ | $c < 0$ |
| Liénard Sigmoid | $\dot{x}_1 = x_2$ | $\dot{x}_2 = -(1/(1 + e^{-ax_1}) - 0.5) - (b + x_1^2)x_2$ | $x_1, x_2 \in [-1.5, 1.5]$ | $a \in [0, 1], b \in [-1, 1]$ | $b < 0$ |
| Van der Pol | $\dot{x}_1 = x_2$ | $\dot{x}_2 = \mu x_2 - x_1 - x_1^2 x_2$ | $x_1, x_2 \in [-3, 3]$ | $\mu \in [-1, 1]$ | $\mu > 0$ |
| BZ Reaction | $\dot{x}_1 = a - x_1 - \frac{4x_1 x_2}{1 + x_1^2}$ | $\dot{x}_2 = bx_1 \left(1 - \frac{x_2}{1 + x_1^2}\right)$ | $x_1, x_2 \in [0, 10]$ | $a \in [2, 19], b \in [2, 6]$ | $b < \frac{3a}{5} - \frac{25}{a}$ |
| Selkov | $\dot{x}_1 = x_1 + ax_2 + x_1^2 x_2$ | $\dot{x}_2 = b - ax_2 - x_1^2 x_2$ | $x_1, x_2 \in [0, 3]$ | $a \in [0.01, 0.11], b \in [0.02, 1.2]$ | $b^{-2} > 2 - 4a \pm \sqrt{1 - 8a}$ |

*Augmented SO* system, which is an instance of a simple oscillator (SO) that was transformed by a random initialization of a neural spline flows Durkan et al. (2019a). All of these systems change behavior from node attractors to limit cycles, i.e. undergo a Hopf bifurcation, at known parametric settings, making them particularly relevant for our application.

See Figure 8 for randomly sampled systems, their behavior and the fitting quality of SPE.

## B.2   SIMULATING SPARSE DATA

The most straightforward method to draw sparse data from each system is to draw points uniformly from the phase-space. However, for high-dimensional systems this runs the risk of under-sampling the region of the invariant set, observations which we require in order to make any meaningful prediction. Moreover, this does not accurately portray how we expect data to be collected in real scenarios. Instead, we assume that each point of the data is a point on a random trajectory, with initial conditions coming from a uniform distribution on pre-specified ranges of the phase-space. Additionally, we assume that each observed pair $(x_i, \dot{x}_i)$ has independently traversed a different amount of time on its respective trajectory. The data sampling process can be summarized through the following steps:

Figure 9: The effects of changing the maximum simulation sampling time $T_{\max}$. When $T_{\max}$ is small (e.g. $T_{\max} = 0.5$; left), most of the observations are transients. On the other hand, when $T_{\max}$ is large (e.g. $T_{\max} = 5$; right), most of the observations are on the invariant set.

1. For each $i \in \{1, \cdots, N\}$, draw the initial conditions $x_i^{(0)}$ uniformly from a pre-specified portion of space

2. Sample the amount of time $t_i$ uniformly in the range $[T_{\min}, T_{\max}]$

3. Simulate a trajectory beginning at $x_i^{(0)}$ to time $t_i$. The point $x_i$ reached on the trajectory in this manner is defined as the position, and $\dot{x}_i = f(x_i)$ as the velocity

4. Optionally, Gaussian noise with standard deviation $\sigma$ is then added to $\dot{x}_i$

Usually we set the parameter $T_{\min} = 0$, except when sampling from the repressilator, where it is set to $T_{\min} = 3$. $T_{\max}$ is a hyperparameter that effectively controls how much samples tend to concentrate around the invariant set. When $T_{\max}$ is close to 0, then the pairs $(x_i, \dot{x}_i)$ revert back to being uniformly sampled from phase-space.

An example of the effect of different values for $T_{\max}$ can be seen in Figure 9. When $T_{\max}$ is small, the observations mostly include transients of the dynamics, in which case not much can be estimated with regards to the invariant set of the system. As $T_{\max}$ increases, more points aggregate near the invariant set. For our 2D simulations we use $T_{\max} = 3$, whereas for the repressilator system we used $T_{\max} = 10$.

All trajectories are simulated with the Runka-Kutta method (RK4) and a step size of $\Delta t = 0.01$.

### B.3 REPRESSILATOR

The repressilator system is governed by six coupled differential equations, structured as three instances of the following gene-regulation model (Elowitz & Leibler, 2000):

$$\dot{m}_i = -m_i + \frac{\alpha}{1 + p_j^n} + \alpha_0,$$

$$\dot{p}_i = -\beta(p_i - m_i),$$

where $m_i$ and $p_i$ denote the mRNA and protein concentrations for gene $i$, corresponding to **LacI, TetR, and $\lambda$ cI**, with cyclic inhibition: $i = $ LacI, TetR, cI and $j = $ cI, LacI, TetR. The parameters $\alpha_0$ (basal transcription), $\alpha$ (maximal transcription), $\beta$ (protein/mRNA degradation ratio), and $n$ (Hill coefficient) define the system's regulatory dynamics.

We analyze trajectories across $\alpha \in (0, 30)$, $\beta \in (0, 10)$, with $\alpha_0 = 0.2, n = 2$. A supercritical Hopf bifurcation emerges at critical values of $\beta$ (Verdugo, 2018b):

$$\beta_1 = \frac{3A^2 - 4A - 8}{4A + 8} + \frac{A\sqrt{9A^2 - 24A - 48}}{4A + 8},$$

$$\beta_2 = \frac{3A^2 - 4A - 8}{4A + 8} - \frac{A\sqrt{9A^2 - 24A - 48}}{4A + 8},$$

where:

$$A = \frac{-\alpha n \hat{p}^{(n-1)}}{(1 + \hat{p}^n)^2}, \quad \hat{p} = \frac{\alpha}{1 + \hat{p}^n} + \alpha_0 \tag{15}$$

These are the boundaries plotted in Figure 4 (top).

To simulate instances of the repressilator, we use the same sampling procedure as described in Appendix B.2, with $T_{\min} = 3$ and $T_{\max} = 10$. In this context we use $T_{\min} = 3$ to ensure that when the samples are projected to lower than six dimensions they still define a vector field with non-intersecting trajectories. Furthermore, even though we used $T_{\max} = 10$, a larger value than in the other simulated systems, we found that the observed points do not all lie on the invariant set of the repressilator (as can be seen in Figure 4).

## C    BASELINE DETAILS

### C.1    BASELINES IN INVARIANT SET LOCALIZATION

In Section 4.1 we showcase the ability of SPE at estimating the position of limit cycles from observations. Few existing approaches to do so exist, and those that do rely on data which is dense on a regular grid. However, the quality of the fit can be evaluated compared to other methods for estimating the underlying ODE. Accordingly, we compared SPE to a number of simple baselines:

- **kNN**, a simple interpolator which uses the $k$-nearest observed positions in order to estimate the velocity. In our experiments, we found that $k = 5$ neighbors approximates clean systems fairly well, and used that setting for all of our comparisons.
- **SINDy (Brunton et al., 2016; de Silva et al., 2020)** is an established method for estimating vector fields from empirical observations. SINDy uses sparse linear regression with a fixed library of functions to estimate the velocities of the system. For our evaluations we use the `pysindy` package (de Silva et al., 2020) and consider a function library with 3rd order polynomials (SINDy-poly) or comprised of 10 Fourier features (SINDy-fourier). All other hyperparameters used were those that are the default in the `pysindy` package.
- **MLP** is a simple multi-layer perceptron with 2 hidden layers, with a width of 128 and SiLU activations (Hendrycks & Gimpel, 2016). These MLPs were trained to estimate the velocities from the positions using an MSE loss over the predicted velocities, with the Adam optimizer (Kingma, 2014), for 5000 (full-batch) iterations. We used a learning rate of 0.001 and a weight decay of 0.001. These hyperparameters were chosen as MLPs trained in such a manner accurately approximate clean systems with $N > 250$ observations.

All of these methods form as alternatives for approximating the underlying ODE from sparse observations. However, unlike SPE, these methods cannot directly localize the invariant set of the observed (or even, fitted) system. Even when the fit is very accurate, determining where and what kind of invariant set was learned is a very difficult problem. Instead, we opt to compare the long-term performance of SPE against all of these baselines, fully described in the next section.

### C.2    EVALUATION OF INVARIANT SET LOCALIZATION

To quantitatively evaluate the localization of the invariant set, for SPE and the other baselines considered, we simulate long trajectories from fixed initial conditions in both the true (hidden) system and in the fitted systems. When the initial conditions are sampled from some distribution, these long trajectories give an effective stationary distribution of points along the invariant set of the considered systems. Accordingly, our goodness-of-fit measure is the Wasserstein-2 ($W_2$) distance between effective stationary distributions of the hidden governing equations and the fitted ODEs.

For the evaluations in Figure 3 (left), we randomly sampled 1000 systems which contain limit cycle behaviors. Systems were sampled uniformly from the SO, Liénard Polyonmial, Liénard Sigmoid, Van der Pol, BZ Reaction and Selkov families of systems (Appendix B.1), with order parameters uniformly sampled from the ranges corresponding to limit cycle behaviors. For each system, different numbers $N$ of positions $x_i$ and velocities $\dot{x}_i$ were sampled according to the procedure described in Appendix B.2, with $T_{\max} = 3$. Additive Gaussian noise was then added to these velocities with a

standard deviation of $\sigma = \mu/\text{SNR}$ where $\mu = \frac{1}{N}\sum_i \|\dot{x}_i\|$ and a prespecified signal-to-noise ration (SNR). Adding noise in this manner allows for direct control of the SNR of the observed velocities between different families of systems, whose phase spaces and velocities are defined on a range of sizes. Each of the methods, SPE and the baselines described in Appendix C.1, were then fitted to all $N$ simulated positions $x_i$ and noisy velocities $\dot{x}_i$ in order to approximate the underlying ODE.

Each method was then evaluated by comparing their effective stationary distribution to that of the ground-truth system. To do so, $M = 1000$ initial conditions were sampled from the range of initial conditions of the corresponding family of systems (see Appendix B.1). The same initial conditions were used to generate long trajectories under the true system and the estimated ODE, and the $W_2$ distance was then calculated between the last time-points for the true system and the fitted ODE. The trajectories were integrated using RK4, with a step size of $\Delta t = 0.01$, for $T_{\text{eval}} = 100$ which we found is long enough for the true system to always be close to its invariant set. To calculate the $W_2$ distance we utilize the Python Optimal Package from Flamary et al. 2021.

## C.3 BASELINES IN CLASSIFICATION

To benchmark SPE in the classification of dynamical systems, we turn to datasets previously considered by Moriel et al. 2023 and compare to the same methods. For all of these baselines, we use exactly the same settings as described in Section 7.4 of Moriel et al. 2023. Below we provide a brief summary of each of these methods:

- **TWA** (Moriel et al., 2023), a convolutional NN trained to classify between point and periodic dynamics on a dense grid. Instead of training on the input vectors directly, TWA predicts the class of the system using an angular representation of each vector in each position. TWA is trained on entirely simulated data, which is generated by sampling the order parameters from the SO family of systems. These are then augmented using neural spline flows (Durkan et al., 2019b), INNs. Augmenting in this way ensures that the same invariant structure is kept between the originally sampled SO system and the augmented version. TWA was then trained on 10,000 of these randomly augmented systems.

- **Critical Points**, is based off of an algorithm devised by Helman & Hesselink 1989 which detects and characterizes critical points from a representation of an autoencoder. A system is considered periodic under CriticalPoints if at least one detected point is a repeler, and is otherwise classified as an attractor.

- **Phase2vec** is a convolutional autoencoder trained to reconstruct 10,000 training systems defined by sparse polynomial equations. The encoder maps to a 100-dimensional latent space using a single layer. The output of the decoder are the predicted coefficients of the parameters, which are multiplied by the polynomial function library in order to reconstruct the original vector field. The latent representations of the autoencoder then form features for an additional linear classifier, which was trained for classification using the same dataset as TWA.

- **Autoencoder** is a convolutional autoencoder with a bottleneck of 10 latent dimensions, trained to directly reconstruct the vectors generated as in Phase2vec. Both encoder and decoder have three convolutional blocks with a stride of $2 \times 2$ in order to reduce the resolution of the input during training. After training, similarly to Phase2vec, the latent representations are turned into features for a linear classifier fitted to the same data as TWA.

Beyond accuracy, we also compared the F1, precision and recall scores of SPE versus each of the above methods. These comparisons are summarized in Table 4 and Table 5. In all families of systems except SO and Aug. SO, SPE most consistently achieves the highest scores.

## C.4 EVALUATION IN SPARSE CLASSIFICATION

To evaluate the classification performance of SPE on sparse and noisy data, we simulated systems in the same manner as in Appendix C.2. We randomly sampled 1000 systems from the SO, Liénard Polyonmial, Liénard Sigmoid, Van der Pol, BZ Reaction and Selʼkov families of systems. For each system, the order parameters were uniformly sampled from their corresponding ranges (see Appendix B.1), which correspond to either node attractors or cyclic attractors. For each system, $N$

Table 4: **Comparison of the F1/Precision/Recall scores for all methods on the classification task from Moriel et al. 2023 (continued in Table 5).** The highest numbers for each column and score are shown in bold.

| | SO | Aug. SO | Lienard Poly | Lineard Sigmoid |
|---|---|---|---|---|
| SPE | 90±1/**100±0**/82±2 | 79±1/88±1/71±2 | **96±1**/100±1/**92±2** | **93±1**/88±2/**98±1** |
| TWA | **97±1**/97±2/97±3 | **93±1**/**92±2**/94±2 | 85±22/**100±0**/78±23 | 90±13/**100±1**/84±17 |
| Phase2Vec | 62±17/**100±1**/47±18 | 60±10/**92±2**/46±11 | 7±22/10±29/8±25 | 0±0/0±0/0±0 |
| Autoencoder | 95±2/93±5/**99±3** | 87±1/84±3/92±4 | 55±37/54±37/64±43 | 84±18/100±1/75±21 |
| Critical Points | 17/87/9 | 34/67/23 | 59/96/42 | 7/**100**/4 |

Table 5: Continuation of Table 4.

| | Van der Pol | BZ reaction | Selkov |
|---|---|---|---|
| SPE | **97±3**/95±5/**100±0** | **87±2**/100±0/77±3 | **66±2**/98±1/50±2 |
| TWA | 80±26/96±20/71±25 | 85±7/78±11/**96±9** | 49±10/**98±5**/34±11 |
| Phase2Vec | 4±12/20±40/3±8 | 0±0/0±0/0±0 | 0±0/0±0/0±0 |
| Autoencoder | 74±31/89±26/71±34 | 81±17/87±13/84±24 | 3±8/12±32/2±5 |
| Critical Points | 33/**98**/20 | 87/79/**96** | 15/62/9 |

positions $x_i$ and velocities $\dot{x}_i$ were sampled according to the procedure described in Appendix B.2, with $T_{\max} = 3$. Additive Gaussian noise was then added to these velocities with a standard deviation of $\sigma = \mu/\text{SNR}$ where $\mu = \frac{1}{N}\sum_i \|\dot{x}_i\|$ and a prespecified signal-to-noise ration (SNR). The task was then to classify the set of pairs $\{(x_i, \dot{x}_i)\}_{i=1}^{N}$ either as a node attractor or a limit cycle.

## D  IDENTIFICATION OF THE CELL CYCLE TRAJECTORY FROM SINGLE-CELL RNA-SEQUENCING AND RNA VELOCITY DATA OF PROLIFERATING CELLS

We analyzed single-cell RNA-sequencing data from U2OS cell line that was integrated with the fluorescent ubiquitination-based cell cycle indicator (FUCCI) in order to investigate the spatiotemporal dynamics of human cell cycle expression (data collected in (Mahdessian et al., 2021)). The original dataset included 1,152 cells characterized by the expression of 58,884 genes. After preprocessing, following the protocol in Zheng et al. (2023) — which involved gene and cell filtering, normalization, log transformation, and the selection of highly variable genes — 1,021 cells remained, described by 2000 genes.

To infer local measures of cellular dynamics in the high-dimensional gene expression space, we used scVelo (Bergen et al., 2020) which computes RNA velocity vectors from spliced and unspliced gene expression counts of a cell population. scVelo models the transcriptional dynamics of each gene using a dynamical model of transcription, splicing, and degradation. It applies an Expectation-Maximization algorithm to estimate gene-specific kinetic parameters and to infer the latent transcriptional state of each cell.

The gene expression data and the inferred velocities were projected into a 100-dimensional space using principal component analysis (PCA). We chose to project the data into a 100-dimensional space in order to reduce the computational load of our method. Streamlines of the resulting velocity field are visualized in Figure 5, left. Independently of the RNA velocities, we computed S-phase and G2/M-phase scores per cell by aggregating the expression of marker genes for these phases, as defined by Tirosh et al. (2016), using the function "score_genes_cell_cycle" from scVelo package (Bergen et al., 2020). The color in Figure 5, left and middle, represents the fractional component of the S-phase score relative to the combined S and G2/M scores.

We then applied our method to fit the RNA velocities, embedded in the 100-dimensional PCA space, to a 100-dimensional cyclic oscillator prototype, as defined in Equation (4) of Section 4.3. This prototype oscillates in the first two dimensions and decays exponentially in the others. The fitting parameters were as follows: angular speed $\omega = 5$, radius $a = 1$, three blocks as defined in Appendix A, $K = 3$ frequencies per layer, weight decay of `wd` $= 1e^{-3}$, projection regularization set to `proj_reg` $= e^{-2}$, determinant regularization of 0, a learning rate of `lr` $= 5e^{-4}$, and 1,000 iterations.

Given the fitted coordinates of the cycle attractor in the 100-dimensional PCA space, we inverted the PCA transformation to reconstruct the corresponding predicted gene expression. In Figure 5 (right), we plot representative marker genes for both the S-phase (top half of the heatmap) and G2/M-phase (bottom half) taken from Tirosh et al. 2016, showing distinct oscillatory patterns consistent with cell cycle progression.

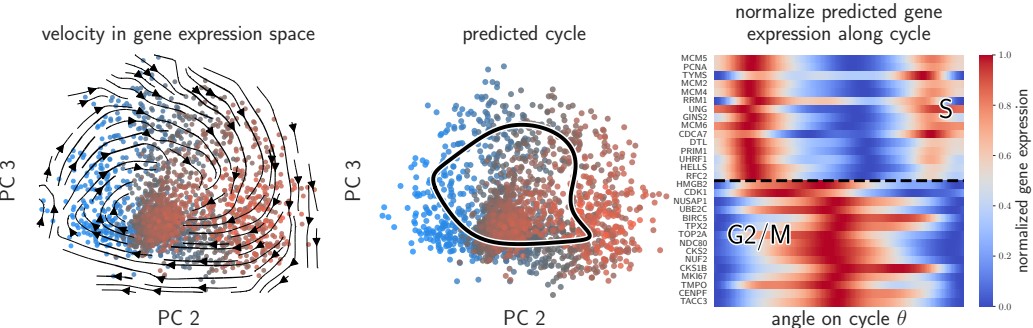

Figure 10: **Fitting of the cell cycle in human fibroblasts (Riba et al., 2022). Left:** gene expression velocity of cells undergoing proliferation in PCA space, as in Figure 5. PCs 2 and 3 are shown as the cycle is more apparent in this projection. **Middle:** the SPE localization of the cycle attractor. **Right:** normalized gene expressions along the cycle attractor as predicted by SPE. The heatmap depicts a set of common marker genes for the S (top of the heatmap) and G2/M (bottom of the heatmap) phases.

These qualitative results are not unique to the FUCCI dataset. Using the same preprocessing, we fit an alternative dataset of human fibroblast cells during their cell cycle (Riba et al., 2022) using 10 PCs, which results in qualitatively similar performance, as can be seen in Figure 10. 10 PCs were used in this dataset to reduce computational load.

## E  THEORETICAL TREATMENT

As a proof of correctness for the use of the equivalence loss, we will consider small perturbations to the true underlying field and the implications of these perturbations for the equivalence loss. We will then show how our diffeomorphisms imply that these perturbations are $C^1$-close to the original field (Guckenheimer & Holmes, 2013). This, together with an assumption regarding the stability of the underlying field, will demonstrate a correspondence between small equivalence loss and the recovery of the true invariant set.

### E.1  CORRECTNESS OF EQUIVALENCE LOSS

We will analyze the effects of small perturbations of the diffeomorphism from its optimal form:
$$H_\delta(x) = H^\star(x) + \delta H(x) \tag{16}$$
where $H^\star(x)$ is the optimal diffeomorphism under which the prototype $g(y)$ and the vector field $f(x)$ are equivalent, $H(x)$ is a diffeomorphism that perturbs $H_\delta(x)$ from the optimum, and $\delta > 0$ is a small positive constant that represents the magnitude of the perturbation. As described, we will look at the case when $f(x)$ and $g(y)$ are smoothly equivalent, meaning:
$$\partial_x H^\star(x) f(x) = g(H^\star(x)) \Leftrightarrow f \overset{H^\star}{\sim} g \tag{17}$$
Furthermore, we will analyze an unnormalized form of the equivalence loss from Equation (2):
$$\mathbb{E}_x \left[ \|\partial_x H_\delta(x) f(x) - g(H_\delta(x))\|^2 \right] \tag{18}$$

Under our weak perturbations, we have:
$$\partial_x H_\delta(x) f(x) = \partial_x H^\star(x) f(x) + \delta \partial_x H(x) f(x) \tag{19}$$
$$= g(H^\star(x)) + \delta \partial_x H(x) f(x) \tag{20}$$

We will further assume that $g \in C^2$ on the data domain, i.e. that it has bounded second derivatives. Then, approximating $g(H_\delta(x))$ with its Taylor expansion around $H^\star(x)$:

$$g\left(H_\delta(x)\right) = g\left(H^\star(x) + \delta H(x)\right) \tag{21}$$

$$= g(H^\star(x)) + \delta \cdot \partial_x g(H^\star(x)) H(x) + \mathcal{O}(\delta^2) \tag{22}$$

Under these conditions, the unnormalized equivalence loss is equal to:

$$\mathbb{E}_x\left[\|\left[\partial_x H_\delta(x)\right] f(x) - g\left(H_\delta(x)\right)\|^2\right] = \mathbb{E}_x\left[\|\delta\left(\left[\partial_x H(x)\right] f(x) - \left[\partial_x g(H^\star(x))\right] H(x)\right) + \mathcal{O}(\delta^2)\|^2\right] \tag{23}$$

$$= \delta^2 \cdot \mathbb{E}_x\left[\|\left[\partial_x H(x)\right] f(x) - \left[\partial_x g(H^\star(x))\right] H(x) + \mathcal{O}(\delta)\|^2\right] \tag{24}$$

So, our equivalence scales together with the size of the perturbations. In other words, optimizing diffeomorphisms to reduce the equivalence loss corresponds with optimizing the diffeomorphism to remove the effects of the added perturbation, $\delta H(x)$.

### E.2 IMPLICATIONS REGARDING TRUE INVARIANT SET

In the above, we showed that the equivalence loss shrinks as the size of the perturbation from the optimal diffeomorphism becomes smaller. This was true for any perturbation from the optimum, but does not necessarily mean on its own that the fitted invariant set is recapitulated. We will now move our attention to this matter.

To be able to faithfully reconstruct the true invariant set, we will only consider vector fields which are structurally stable (Guckenheimer & Holmes, 2013), in the sense that their invariant sets are retained under small $C^1$-perturbations. In our scenario, the vector field modeled by SPE is a function of both the prototype and the fitted diffeomorphism, and is given by:

$$f_H(x) = \left[\partial_x H_\delta(x)\right]^{-1} g\left(H_\delta(x)\right) \tag{25}$$

$$= \left[\partial_x H^\star(x) + \delta \partial_x H(x)\right]^{-1} g\left(H_\delta(x)\right) \tag{26}$$

$$= \left[\partial_x H^\star(x) + \delta \partial_x H(x)\right]^{-1} \left[g(H^\star(x)) + \delta \partial_x g(H^\star(x)) H(x) + \mathcal{O}(\delta^2)\right] \tag{27}$$

$$= \left[I + \delta\left(\partial_x H^\star(x)\right)^{-1} \partial_x H(x)\right]^{-1} \left[f(x) + \delta\left(\partial_x H^\star(x)\right)^{-1} \partial_x g(H^\star(x)) H(x) + \mathcal{O}(\delta^2)\right] \tag{28}$$

where the final step is due to the equality (through smooth equivalence):

$$f \stackrel{H^\star}{\sim} g \Leftrightarrow f(x) = \left[\partial_x H^\star(x)\right]^{-1} g(H^\star(x)) \tag{29}$$

The above can be further expanded through the Neuman series when the eigenvalues of $\left[\partial_x H^\star(x)\right]^{-1} \partial_x H(x)$ are bounded and $\delta$ is small enough such that $\delta|\lambda_i(\left[\partial_x H^\star(x)\right]^{-1} \partial_x H(x))| < 1$ with $\lambda_i(A)$ the eigenvalues of $A$ (Petersen et al., 2008):

$$\left(I + \delta\left(\partial_x H^\star(x)\right)^{-1} \partial_x H(x)\right)^{-1} = \sum_{\ell=0}^{\infty}(-\delta)^\ell \left[\left(\partial_x H^\star(x)\right)^{-1} \partial_x H(x)\right]^\ell \tag{30}$$

$$= I + \sum_{\ell=1}^{\infty}(-\delta)^\ell \left[\left(\partial_x H^\star(x)\right)^{-1} \partial_x H(x)\right]^\ell \tag{31}$$

$$= I - \delta\left(\partial_x H^\star(x)\right)^{-1} \partial_x H(x) + \mathcal{O}(\delta^2) \tag{32}$$

Plugging this in, the perturbed vector field is given by:

$$f_H(x) = \left(I - \delta\left(\partial_x H^\star(x)\right)^{-1} \partial_x H(x) + \mathcal{O}(\delta^2)\right) \left[f(x) + \delta\left(\partial_x H^\star(x)\right)^{-1} \partial_x g(H^\star(x)) H(x) + \mathcal{O}(\delta^2)\right] \tag{33}$$

$$= f(x) + \delta \cdot \left(\partial_x H^\star(x)\right)^{-1} \left[\partial_x g(H^\star(x)) H(x) - \partial_x H(x) f(x)\right] + \mathcal{O}(\delta^2) \tag{34}$$

The perturbation $\epsilon(\delta) = f(x) - f_H(x)$ from $f(x)$ is thus equal to:

$$\epsilon(\delta) = \delta \cdot (\partial_x H^\star(x))^{-1} \left[ \partial_x g(H^\star(x)) H(x) - \partial_x H(x) f(x) \right] + \mathcal{O}(\delta^2) \tag{35}$$

All elements of this perturbation are scaled by $\delta$. Thus, when all elements of the Jacobian of both $H(x)$, $H^\star(x)$ and $f(x)$ are bounded in the region of the data, then $f_H(x)$ is a $C^0$ perturbation of $f(x)$ in the limit $\delta \to 0$, the range where our equivalence loss is low. In addition to the above, $C^1$ perturbations, defined by the magnitude of the partial derivatives $\partial \epsilon(\delta)/\partial x_i$, can also be guaranteed under specific conditions. In particular, the partial derivatives of the perturbation $\epsilon(\delta)$ with respect to the inputs $x_i$ will depend on all partial derivatives up to order 2 and their products of both the diffeomorphism, $\partial^2 H(x)/\partial x_i \partial x_j$, the optimal diffeomorphism, the prototype, and the underlying vector field. Therefore, if all of these derivatives, up to the second order, are bounded from below and above by a constant, then $C^1$ closeness will also be guaranteed in the limit of $\delta \to 0$.

Under these conditions then, as shown above, $f(x)$ and $f_H(x)$ are $C^1$-close at $\delta \to 0$. Since $f(x)$ was assumed to be structurally stable, this means that $f(x)$ and $f_H(x)$ share the same invariant sets (Guckenheimer & Holmes, 2013).

**To summarize,** consider the following constraints:

- The prototype, $g(y)$, is smoothly equivalent to the underlying vector field, $f(x)$. Denote by $H^\star(x)$ a diffeomorphism such that $f \overset{H^\star}{\sim} g$.

- In the compact space containing all observed data points, the partial derivatives of up to order 2 for $f(x)$, $g(y)$, $H^\star(x)$ and $H(x)$ are bounded from above and below by a constant.

- In the same compact space, the vector field $f(x)$ is structurally stable (and thus, $g(y)$ is also structurally stable).

Under the above conditions, we have:

1. Minimizing the equivalence loss defined in Equation (2) pushes to decrease the amount of perturbation from the optimal diffeomorphism, and subsequently the perturbation from the true vector field.

2. Under small perturbations, the true vector field and the modeled fields are $C^1$ close.

3. The above, together with the assumption that $f(x)$ is structurally stable, means that when the equivalence loss is low, the true invariant sets of the dynamics are faithfully estimated.

# F  FURTHER EXPERIMENTS

## F.1  MULTIPLE BASINS OF ATTRACTION

While this work mostly focused on prototypes with only a single basin of attraction, such as a node attractor or limit cycle, SPE can be used when there are multiple basins of attraction.

As a simple example, we looked at synthetic dynamics with multiple fixed points in a high-dimensional space. The governing equations for these dynamics were based on a mixture of experts decomposition of space, where each expert is a basin of attraction. Specifically, the governing equations of the dynamics were defined as:

$$f(x) = \sum_{k=1}^{K} r_k(x) \, \varphi_k(x - c_k) \tag{36}$$

$$r_k(x) = \frac{\exp\left[-\frac{1}{2s^2}\|x - c_k\|^2\right]}{\sum_{k'} \exp\left[-\frac{1}{2s^2}\|x - c_{k'}\|^2\right]} \tag{37}$$

where $\varphi_k(\cdot)$ are the local dynamics around the center $c_k \in \mathbb{R}^d$, $s$ determines the strength with which one basin of attraction transitions to another, and $K$ is the number of basins of attraction.

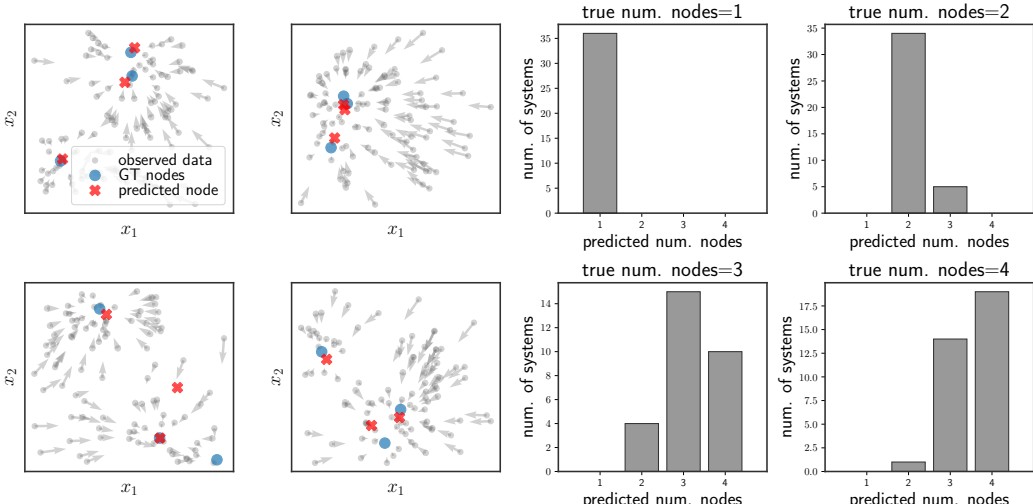

Figure 11: **Classifying and localizing invariant sets in systems with multiple basins of attraction.**
**Left:** 4 examples of fitting SPE to a system with multiple basins of attraction in 3 dimensions. In
each example, the positions of the node attractors was randomly sampled (these positions are depicted
by the blue circles in the plot), and 100 position-velocity pairs were then sampled using the scheme
described in Appendix B.2. SPE was then used to match between this data and a fixed prototype with
3 basins of attraction. The estimated positions of the attractors according to SPE is depicted by the
red crosses. We find that SPE frequently matches the positions of the nodes. **Right:** results in the
classification of the number of nodes. Each bar plot shows how SPE categorizes systems with a set
number of nodes, for instance the top left plot shows the performance only on systems that had one
node attractor. In each of these cases, the main mode predicted by SPE aligns with the true number
of nodes in the system.

Under this setting, we considered systems whose centers were sampled uniformly from the range
$[-0.9, 0.9]^d$, all of which were node attractors:

$$\varphi_k(y) = -y \tag{38}$$

Finally, the number of attractors was randomly chosen to be between 1 and 4. We then fit SPE with 4
different prototypes, for each number of possible basins of attraction. The positions of the nodes in
the prototypes were fixed and distinct from the true underlying system, so that the task was both the
prediction of the location of the nodes as well as the estimation of their number. Results can be seen
in Figure 11 for systems with 100 points.

Throughout the experiments shown in Figure 11, INNs with 2 blocks of width 3 were used, together
with 5 Householder transformations. The networks were trained for 1500 iterations, with a learn-
ing rate of $1e^{-3}$, weight decay of 0.1, with all other regularization coefficients equal to 0. The
classification experiment was carried out in 5D.

### F.2 MISMATCH BETWEEN PROTOTYPE AND UNDERLYING SYSTEM

SPE uses a soft notion of equivalence to match between the prototypes and the data. Thus, even
when there is a mismatch between the prototype and the underlying system, we expect SPE to
capture the *dominant behavior* of the system. To test this, we looked at the families of systems from
Appendix B.1, and simulated data from oscillatory systems. As the limit cycle of these systems is
bounded to a finite region, the dominant behavior of the system as a whole is determined by the
orientation of the flow - either clockwise or counter-clockwise. To emulate a mismatch between the
chosen prototype and the underlying behavior, we fit SPE using only node attractor prototypes, as
opposed to the true underlying oscillatory behavior, using the same prototypes from Equation (3),
with $a = -0.25$ and $\omega = \pm 0.5$. Figure 12 demonstrates that even when there is a mismatch between
the underlying dynamics and the prototype, SPE is able to capture the dominant behavior of the

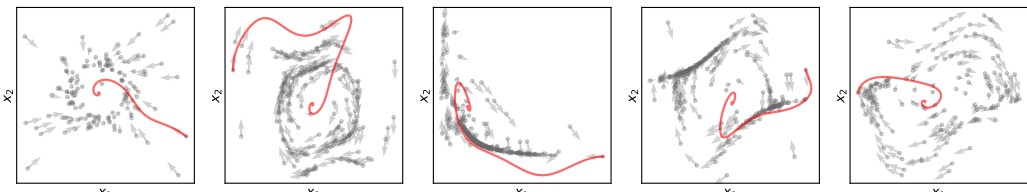

Figure 12: **Direction of angular velocity is captured, even when there is a mismatch between the prototype and the true underlying system.** Randomly sampled systems and examples of fitting SPE to an inherently oscillatory process using a node attractor prototype. Despite this mismatch between underlying behavior and prototype, SPE is still able to characterize the dominant behavior of the system, mainly the direction of the angular velocity and the focal point of the limit cycle.

system - that is, the orientation of the flow is matched. Quantitatively, we find that the average accuracy when trying to predict the orientation of the SO family of systems was 98%, averaged over 100 trials using the same hyperparameters described in Appendix A.7.

This demonstration illustrates how SPE might be used, even when the prototype does not fully match the ground-truth behavior. This matches scientific scenarios, where there is usually some domain expertise as to the dominant behavior of an observed system, but this knowledge may not capture all subtleties of the empirically collected data. In such scenarios, it can still be beneficial to match between an analytical model (i.e. the prototype) and the data for scientific discovery.

### F.3 LIMIT CYCLE PROTOTYPES FROM ALTERNATIVE OSCILLATORS

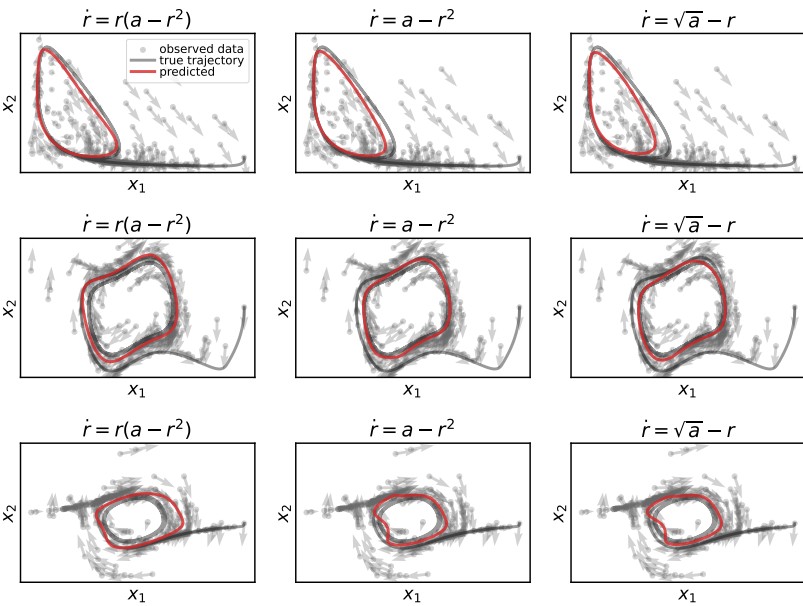

Figure 13: **Estimation of limit cycles using different oscillator prototypes.** The three rows correspond to different possible cycle prototypes. The left-most column corresponds with the prototypes used in the rest of this work, and the middle and right-most columns have decreasing speeds towards the limit cycle in the radial direction. The three prototypes result in similar estimated invariant sets.

Throughout the main text, the form of the prototypes we used were:

$$\dot{r} = r(a - r^2), \qquad \dot{\theta} = \omega \tag{39}$$

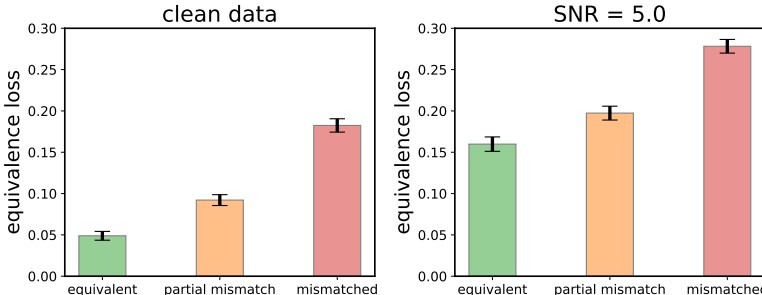

Figure 14: **The equivalence loss can be used as a measure of prototype mismatch,** for both clean (left) or noisy (right) data. The equivalence is averaged over multiple instances of the systems from Table 3, fitted with the node or cycle prototypes, which can either be partially mismatched with the data (when the angular velocity matches, but not the invariant set), or fully mismatched. The error bars represent the standard error of the mean of the average equivalence loss.

where $r$ and the $\theta$ are polar coordinates. We chose this prototype as it is a well-studied form of a supercritical Hopf system. However, SPE is robust to the use of other governing equations which result in limit cycles, as these systems are smoothly equivalent to each other.

Two alternative examples for oscillators which can be used as prototypes are:

$$\dot{r} = a - r^2, \qquad \dot{\theta} = \omega \tag{40}$$

$$\dot{r} = \sqrt{a} - r, \qquad \dot{\theta} = \omega \tag{41}$$

These two systems are qualitatively equivalent to each other and to Equation (39). These systems exhibit different speeds in the radial direction towards the radius of the limit cycle, but otherwise remain qualitatively similar. Using SPE with these prototypes results in very similar estimated invariant sets, as shown in Figure 13.

## F.4 DETECTING PROTOTYPE MISMATCH

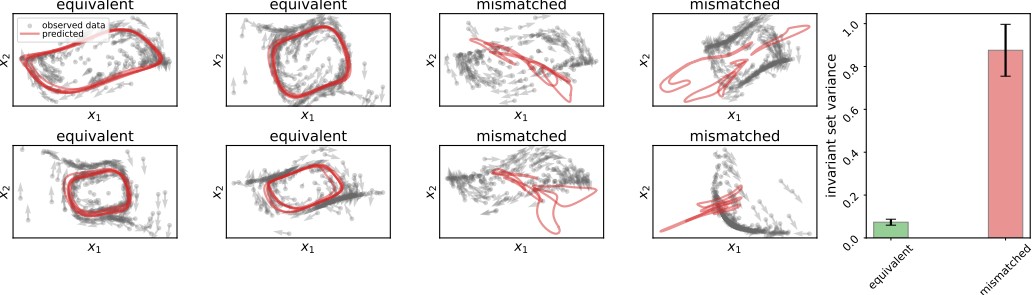

Figure 15: **Effects of mismatch between the true system and the prototype on the estimated invariant set. Left:** when the two systems are equivalent, the location of the estimated invariant set is robust to randomly subsampling the positions. **Middle:** on the other hand, when there is a mismatch between the prototype and system, the diffeomorphism tends to overfit. In such scenarios, for every random subsampling of the data, a dramatically different limit cycle is estimated. **Right:** the variance, average over the whole invariant set, in the estimated positions is enough to differentiate between equivalent and mismatched prototype-true system pairs. The error bars represent the standard error of the mean of the variance.

We expect the equivalence loss from Equation (2) to correspond with the level of mismatch between the true underlying vector field and the prototype. As shown in Table 1, this allows for accurate classification between possible prototypical systems. Beyond this, the loss after optimization can be

used as a measure of the mismatch between prototype and data; To demonstrate this, we examined 300 instances of the simulated systems from Table 3, corresponding to both cycles and node attractors. We then randomly assigned the sign of the angular velocity of the prototype, $\omega$, and radius of the limit cycle $a$ (where negative values correspond to node attractors). In this setting, systems can be either: (1) equivalent (angular velocity and radius match), (2) partially mismatched (angular velocity matches, but radius does not), (3) mismatched (both angular velocity and radius are mismatched relative to the true system). For systems that are equivalent, setting (1), we expect a small loss. For a partial mismatch, which was qualitatively analyzed in Appendix F.2, we expect the loss to be larger than the equivalent systems. Finally, when the two systems are mismatched, we expect the loss to be larger still, accounting for the error in the direction of the vectors for the whole space. This expected trend is what is demonstrated in practice, as shown in Figure 14 (left). The average loss is lowest for systems that are equivalent to each other, and highest for systems that are mismatched. This trend holds for noisy data (SNR = 5), as shown in Figure 14 (right).

Another indication for a possible mismatch between the prototype and the underlying system can be identified by the variance of the estimated invariant set under multiple initializations. When the true system and the prototype are mismatched, we find that the parameterized diffeomorphism overfits to the data, whereas equivalent systems are robust to small changes in the data. In particular, when there is a mismatch, the estimated invariant set changes dramatically under a random subsampling of the data. Examples of this behavior can be seen in Figure 15, where 5 different diffeomorphisms were fit to 5 random splits of the data, in which only 50% of the data was kept in each split. When the prototype and underlying system are equivalent, the fitted invariant set is robust to the particular split (Figure 15, left). On the other hand, when there is a mismatch, the position and shape of the invariant sets change wildly (Figure 15, middle). To quantify this effect, we sampled 300 systems and randomly chose prototypes that are either equivalent or mismatched (as in Figure 14). We then calculated the variance of the positions on the estimated invariant set, averaged over the full limit cycle. Figure 15 (right) shows that this variance is much higher for mismatched prototype-system pairs when compared to those that are equivalent. Thus, we expect that this measure of fitting variance, paired with the equivalence loss, can be used in order to detect out-of-dictionary scenarios, where the prototypes and data do not exhibit the same behavior.

