# OpenReview forum: "Characterizing Nonlinear Dynamics via Smooth Prototype Equivalences"
_ICLR.cc/2026/Conference — Submitted to ICLR 2026_

### Official Review · Reviewer_AQH7 · 2025-10-19

**Soundness:** 3
**Presentation:** 2
**Contribution:** 3
**Rating:** 4
**Confidence:** 2

**Summary:**

This paper introduces a method called Smooth Prototype Equivalences (SPE) for characterizing dynamical systems from sparse, noisy observations without explicit governing equations. SPE uses invertible neural networks to learn smooth diffeomorphisms between observed data and predefined prototype dynamics (e.g., limit cycles or fixed points) to exploiting the systems sharing the same qualitative behavior. The method has two purposes: localizing invariant structures by mapping prototype attractors to data space via the learned inverse transformation, and classifying dynamical regimes by comparing equivalence losses across prototypes. Experiments show SPE outperforms existing baselines on sparse, noisy data and successfully extends to high-dimensional systems, including a 6D gene regulatory network and real 100-dimensional single-cell RNA velocity data where it recovers cell cycle gene expression patterns.

**Strengths:**

(1) Technical contribution with Fourier Feature Coupling layers is a novel idea;
(2) The use of prototypes (e.g., limit cycle, fixed point) provides clear physical and qualitative meaning to the learned mappings;
(3) The method can infer invariant structures and classify system behaviors directly from sparse vector-field data without knowing governing equations;
(4) The method effectively scales to high-dimensional systems and identifies the cell cycle from real single-cell RNA data, which shows practical value in biological science in a data-driven approach.

**Weaknesses:**

(1) Requires predefined prototypes based on domain knowledge in advance, which limits its applicability;
(2) Training multiple invertible neural networks for different prototypes can be expensive, especially for high-dimensional data;
(3) The method only guarantees local equivalence near observed data points.

**Questions:**

(1) Since SPE requires a set of prototype systems (e.g., limit cycles, fixed points), how should users select or construct these prototypes in practice? What happens if the true system exhibits a behavior not covered by the prototype?
(2) Is the learned diffeomorphism $H_\theta$ unique? Could different network initializations produce distinct but equally valid mappings?
(3) Why Fourier features specifically? Have you tried other function approximators like neural ODEs or implicit layers for the coupling?
(4) Given that Koopman operator methods are now also popular tools for data-driven dynamical analysis, how does SPE relate to them? Could these two frameworks complement each other? In your SPE equation (i.e., Eq. 1), the $\partial_x H(x)\cdot \dot{x}$ is equivalent to $dH(x)/dt = \mathcal{L}H(x)$ where $\mathcal{L}$ is the Koopman generator. How would you think of it? Perhaps, considering the Koopman operator framework would help enhance the mathematical formulation of your method. Here is a list of papers you may have interest:

  (a) https://link.springer.com/article/10.1007/s00332-015-9258-5

  (b) https://www.aimsciences.org/article/doi/10.3934/jcd.2015005

  (c) https://epubs.siam.org/doi/10.1137/21M1401243

  (d) https://link.springer.com/article/10.1007/s11071-005-2824-x

  (e) https://pubs.aip.org/aip/cha/article/27/10/103111/151485/Extended-dynamic-mode-decomposition-with

  (f)  https://pubs.aip.org/aip/cha/article-abstract/35/10/103123/3368087/A-data-driven-framework-for-Koopman-semigroup?redirectedFrom=fulltext

---

> ### Author Response · Authors · 2025-11-20
>
> We thank the reviewer for their constructive comments, and their appreciation of the novelty of our technical contributions, the interpretability of the learned mappings, the method’s applicability to sparse realistic data in an equation-free manner, the method’s scalability, as well as its practical value in biological science. We have addressed each point below, and revised the manuscript accordingly.
>
> ---
>
> > _“Requires predefined prototypes based on domain knowledge in advance, which limits its applicability” _
>
> We would like to emphasize that the assumption that a set of potential prototypes is known in advance is rooted in realistic scientific settings. For instance, in computational biology, many of the behaviors empirically observed in single-cell data of multicellular systems such as those that are studied in our work correspond to a limited set of possible behaviors. These include limit cycles which correspond with cells exhibiting periodic behaviors (i.e. cell cycle or circadian rhythm) such as demonstrated in Fig. 5, or point attractors in space which correspond with cells converging at pre-defined end-states (i.e. cell differentiation) [1,2,3]. This setting was the main focus of our work, corresponding to strong prior information on the types of dynamics to expect in the observed data, and we believe has relevance beyond scenarios in biology. In future implementations, these simple, local prototypes could be combined in a single system in order to model more complex behaviors, a point which we have added to the Discussion.
>
> _[1]: Riba et al., "Cell cycle gene regulation dynamics revealed by RNA velocity and deep-learning" (2022)_
>
> _[2]: Gupta et al., "Simulation-based inference of differentiation trajectories from RNA velocity fields” (2022)_
>
> _[3]: Mayer et al., “The tumor microenvironment shows a hierarchy of cell-cell interactions dominated by fibroblasts” (2023)_
>
> ---
>
> > _“Training multiple invertible neural networks for different prototypes can be expensive, especially for high-dimensional data”_
>
> To account for the challenges of high dimensional data, we used narrow blocks (with few Fourier coupling frequencies), together with Affine transformations with low rank; These scale much better with the dimension of the data than standard normalizing flows, using full-rank Affine transformations and MLP Coupling layers, which would require wider networks. However, we agree with the reviewer that in higher dimensional settings, the training of the diffeomorphisms can be challenging. We expect it will be possible in future works to integrate methods from other fields of machine learning (such as computer vision) in order to more efficiently parameterize the required transformations.
>
> ---
>
> > _“Since SPE requires a set of prototype systems (e.g., limit cycles, fixed points), how should users select or construct these prototypes in practice?”_
>
> Using our methodology, the exact details of the governing equations are less important than the topology of the invariant set, as the learned network can correct for differences in the vector field as long as the two have the same invariant set (as demonstrated in Fig. 2), which we view as a strength of our approach.
>
> In our simulated scenarios and in the analysis of real biological data, the expected invariant structures are limit cycles or nodes. As such, we chose a well known system with governing equations that portray both, which is a version of the supercritical Hopf bifurcation (which we called the “simple oscillator”, or SO, in the text) given by the governing equations in polar coordinates: $\dot{r}=r(a-r^2)$ and $\dot{\theta}=\omega$. In our context, the supercritical Hopf bifurcation serves as a useful model for both limiting behaviors, as the governing equations are well known and are easy to calculate.
>
> However, for this kind of data, any governing equations portraying limit cycles would have worked, as they are all in the same equivalence class. For instance, the radial velocity could be changed to $\dot{r}=a-r^2$ or $\dot{r}=\sqrt{a}-r$, and the long-term behavior for these systems would still converge to a limit cycle. Thus, all three systems are valid contenders as prototypes for cyclic data, and can work in our approach. These three parameterizations converge to very similar results, a qualitative experiment which we have now added in Appendix F3 (Fig. 13) of the revised manuscript.
>
> Following this logic, more complex prototypes can be constructed, guided by the behaviors to be modeled in the data. Specifically, as we show in Appendix F1 (Fig. 11), prototypes can also take the form of multiple fixed points, which would coincide with situations when the dynamics converge to one out of K different attractors.

---

> > ### Author Response · Authors · 2025-11-20
> >
> > > _“What happens if the true system exhibits a behavior not covered by the prototype?”_
> >
> > When the behavior underlying the data is different from the chosen prototype, we expect the smooth equivalence loss to be high, and indeed, this allowed us to classify between prototypes. In Appendix F4 (Fig. 14) of the revised manuscript, we show that the average loss of prototypes of different equivalence classes is higher than those in the correct equivalence class. Beyond these, we expect SPE to be used together with explicit domain knowledge for verification, as we have done for scRNA-seq of multicellular systems in Section 4.4 of the manuscript.
> >
> > When the mismatch with the system is only partial, we found that SPE is frequently still able to capture dominant behaviors. For example, as shown in Appendix F2 (Fig. 12), we found that SPE is proficient at classifying between clockwise and counter-clockwise systems using only node attractor prototypes, even when the ground-truth systems exhibited limit cycles. In the same experiment, we saw that the angular velocity and focal point of the data were well captured - i.e. the dominant effects in the data - despite the mismatch between prototype and hidden system.
> >
> > We now relate to this point more explicitly in the Discussion of the revised manuscript.
> >
> > ---
> >
> > > _“Is the learned diffeomorphism H_\theta unique?”_
> >
> > The learned mapping is not necessarily unique, for multiple different reasons, including:
> > 1. As with most neural networks, the construction of the diffeomorphism might not be identifiable, in the sense that multiple different parameter settings correspond to the same functional output. In our scenario, for instance, the scaling of an affine transformation can be negated by one that is deeper in the network, with no effects on the mapping that was learned. Therefore, the parameters of the network are not unique, even when the functionality of the mapping is the same.
> > 2. Because the scenarios we explored involve sparse and noisy data, the learned mapping might qualitatively change between different initializations of the network. To be specific, the model can change its behavior outside the scope of the data, as long as observed points are not negatively impacted.
> >
> >  ---
> >
> > > _“Why Fourier features specifically?”_
> >
> > Many of our design choices were a product of choosing an expressive transformation, under the constraint that the Jacobian-vector-products needed for the loss can be efficiently calculated in closed-form, and also that there is an explicit form for the inverse of the transformation. We elaborate on these points in Section 3.3. That being said, we did experiment with other choices.
> >
> > In our early experiments we tried to use standard MLP coupling layers, as well as neural ODEs, instead of the Fourier feature coupling. However, because the calculation of the Jacobian needed for the loss is nontrivial, and has to rely on autograd methods, we found training was substantially slower. In contrast, the Fourier features have a closed-form expression for the Jacobian, while remaining just as expressive theoretically. Furthermore,  the Fourier features worked well (and particularly better in practice than MLPs) and provided additional benefits, including explicit control over the transformation frequencies, making them our preferred parameterization.
> >
> > We also tried using NODEs as the diffeomorphism, but found that many steps are required in the transformation itself in order to get a faithful estimate of the inverse-transformation, which we heavily use for the localization of the invariant sets. It is also much much slower, especially since the Jacobian needs to be estimated.
> >
> > Nevertheless, we expect that the SPE approach can work with other parameterizations, which could be further optimized in future work.
> >
> > ---
> >
> > > _“Given that Koopman operator methods are now also popular tools for data-driven dynamical analysis, how does SPE relate to them?”_
> >
> > Thank you for the relevant literature, which we have added to the Related Works section. We believe our approach is complementary to, but distinct from parallel work in Koopman analysis. Indeed, following the literature you cite, it can be shown that the two approaches coincide when the observed dynamics are linear. Otherwise, Koopman analysis requires lifting the dynamics to a new measurement space where dynamical equivalence manifests as conjugacy (similarity) of the linearized transition operators. Our approach, on the other hand, can discover this conjugacy in the raw, observed space, which both aids in interpretability and avoids the problem of positing or inferring the measurement functions that are required for linearization in the Koopman case. Overall, we are intrigued by the complementarity of these methods and include a brief note on this subject in the Discussion section.

---

> > > ### Comment · Reviewer_AQH7 · 2025-11-25
> > >
> > > Thank you for addressing my doubts. They look great. I will raise the score and confidence for your effort.

---

### Official Review · Reviewer_1nY5 · 2025-10-31

**Soundness:** 3
**Presentation:** 3
**Contribution:** 3
**Rating:** 6
**Confidence:** 3

**Summary:**

This paper proposes 'Smooth Prototype Equivalences (SPE)', a novel framework for characterizing the long-term behavior of nonlinear dynamical systems from sparse, noisy, and high-dimensional data, a common challenge in the physical and biological sciences.
The core idea is to learn a mapping that 'smoothly' deforms the data space into a known prototype space using Invertible Neural Networks (INNs), rather than directly modeling the complex, unknown dynamics. This mapping aligns the observed data with simple, well-understood 'prototype' dynamics (e.g., stable fixed points, simple limit cycles).
This approach provides two main functions:
	Localization: It pinpoints the shape and location of hidden 'invariant sets' (e.g., limit cycles, fixed points) in the original data space via the learned inverse mapping (H^(-1)).
	Classification: It allows for classifying the system's dynamical regime by comparing the goodness-of-fit (lowest loss) across multiple prototypes.
The authors demonstrate that SPE is robust, outperforming existing techniques, especially in realistic, data-scarce, and noisy scenarios. Moreover, they successfully apply this method to high-dimensional biological data (single-cell gene expression data) to extract the complex 'cell cycle' trajectory.

**Strengths:**

1. Data Efficiency and Robustness: This is the method's strongest point. As shown in Figure 3, it performs far more stably and accurately than other methods (SINDy, MLP), even with very few samples (N=25) and significant noise. This is critical for real-world scientific applications.
2. Interpretability and Localization: Unlike simple 'black-box' classifiers (e.g., TWA), SPE provides the actual shape and location of the invariant set in the data space via H^(-1). This offers scientists deeper insights—not just "what kind?" but also "where and in what shape?"
3. Scalability and Practicality: The method's effectiveness is demonstrated beyond 2D examples, scaling successfully to 6D (Figure 4) and 100D (Figure 5, scRNA-seq) high-dimensional data. The extraction of a biologically meaningful trajectory from 100D real-world cell cycle data is a particularly impressive result.
4. Equation-Free Approach: It can identify the core structure of a dynamical system  without any knowledge of the system's governing equations.

**Weaknesses:**

1.	Dependence on Prototypes: The method relies on the user defining a 'dictionary' of prototypes in advance, based on what dynamics they expect to find. If the true dynamics are of a completely novel form or are not in the dictionary, the classification and localization may fail.
2.	Limitations on Complex Dynamics: As the authors note, this work primarily focuses on simple attractors like stable fixed points or limit cycles. Chaotic systems, characterized by 'strange attractors' with fractal structures, are difficult to map to simple prototypes using smooth equivalence.

**Questions:**

1.	How sensitive are the accuracy and stability of the results to the INN architecture (e.g., number of blocks, number of Fourier features)? Could you share any empirical guidelines for hyperparameter tuning?
2.	A question regarding the construction of the prototype dictionary: How would SPE handle a complex system that contains multiple different dynamical behaviors (e.g., a system with two fixed points and one limit cycle coexisting)?

---

> ### Author Response · Authors · 2025-11-20
>
> We thank the reviewer for their evaluation of our approach being novel, robust, interpretable, equation-free, and scalable, and its practicality and successful application to high-dimensional biological data. We address each of the reviewer’s points below and have revised the manuscript accordingly.
>
> ---
>
> > _“The method relies on the user defining a 'dictionary' of prototypes in advance, based on what dynamics they expect to find...”_
>
> While it is true that SPE does not capture the full dynamics of systems outside the pre-specified prototypes, in practice we found that it frequently captures dominant behaviors even when there is a mismatch. For example, as shown in Appendix E2 (Fig. 12, Appendix F2 in the revised version), we found that SPE is proficient at classifying between clockwise and counter-clockwise systems using only node attractor prototypes, even when the ground-truth systems exhibited limit cycles. In the same experiment, we saw that the angular velocity and focal point of the data were well captured - i.e. the dominant effects in the data - despite the mismatch between prototype and hidden system (Appendix F2).
>
> Moreover, the assumption that a set of potential behaviors is expected in advance has relevant scientific implications. As a specific example, many of the behaviors empirically observed in multicellular systems such as those probed in our work correspond to a limited set of possible behaviors. These include limit cycles for cells exhibiting periodic behaviors (i.e. cell cycle or circadian rhythm) such as demonstrated in Fig. 5, or point attractors in space which correspond with cells converging at pre-defined end-states (i.e. cell differentiation) [1,2,3]. While our illustrative example came from the world of biology, we believe this approach to be applicable, and relevant, to scenarios beyond those in biology. In particular, we expect that combinations of locally simple dynamics, such as nodes and cycles, can be combined in the future to model more complex dynamical systems, in high-dimensional spaces.
>
> _[1]: Riba et al., "Cell cycle gene regulation dynamics revealed by RNA velocity and deep-learning" (2022)_
>
> _[2]: Gupta et al., "Simulation-based inference of differentiation trajectories from RNA velocity fields” (2022)_
>
> _[3]: Mayer et al., “The tumor microenvironment shows a hierarchy of cell-cell interactions dominated by fibroblasts” (2023)_
>
> ---
>
> > _“... Chaotic systems, characterized by 'strange attractors' with fractal structures, are difficult to map to simple prototypes using smooth equivalence.”_
>
> We would like to emphasize, as briefly touched upon in the noted Discussion, that we believe the learning of chaotic attractors directly from data to be a categorically different setting than our prototype approach, despite being of great interest. To an extent, this is because chaotic systems are provably not smoothly equivalent to each other when they have different fractal dimensions. This makes choosing a “chaos prototype” challenging. The problem of mapping chaotic systems to each other is further exasperated when only noisy, sparse samples of the phase space are observed. Instead, the main focus of our work was the modeling of attractors including fixed points and limit cycles, which also arise in many scientific settings such as computational biology. As such, much of the main text of the manuscript focused on these exact settings, but we also considered systems with multiple fixed points in the original submission, and are now in Appendix F1 (Fig. 11) in the revised manuscript.
>
> ---
>
> > _“How sensitive are the accuracy and stability of the results...?”_
>
> We thank the reviewer for their comment, and we have added  a section to the appendix of the revised manuscript, Appendix A8, focused on sensitivity analysis and empirical guidelines. We generally found that the INN architecture is robust to small changes. For instance, we found that up to 6 Fourier features per-layer was expressive enough for most uses, but that when using many more the networks have a tendency to overfit. We now show this in Appendix A8 (Fig. 6) of the revised manuscript. We have additionally added more ablations of hyperparameters in the same Appendix, which we hope demonstrates the stability of the networks to the choices of different parameter settings.
>
> ---
>
> > _“How would SPE handle a complex system that contains multiple different dynamical behaviors...?”_
>
> This is an important setting, which we have started exploring in the appendix of the submitted version of the manuscript, where we conducted experiments with multiple fixed points. These are in Appendix F1 (Fig. 11) of the revised manuscript, where we show that it is possible to capture the dynamics of multiple fixed points using SPE. We believe that further generalizing these results is an important future step, and have added this point to the Discussion.

---

> > ### Comment · Reviewer_1nY5 · 2025-11-27
> >
> > Thanks for the detailed response and revisions. The added sensitivity analysis (App. A8) and multi–fixed point experiments (App. F1) address my main questions and improve clarity. One remaining request: please make the prototype/dictionary limitation more explicit in the main text, ideally with a simple diagnostic for mismatch/out-of-dictionary cases (e.g., loss threshold/uncertainty) and a brief discussion on how SPE might extend to coexisting heterogeneous behaviors (e.g., fixed points + limit cycle). Overall, the revisions strengthen the paper.

---

> > > ### Author Response · Authors · 2025-12-01
> > >
> > > As per the above request, we have extended the Discussion section to further clarify the limitations of our approach, as well as how SPE might be extended to capture coexisting heterogeneous behaviors and more complex behaviors.
> > >
> > > Additionally, we have added a section to the appendix (Appendix F4) which analyzes the behavior of the loss and the uncertainty of the prediction, showing that they can be used as indicators for mismatch/out-of-dictionary cases. Briefly, in Appendix F4 (Fig. 14) we now demonstrate that the equivalence loss increases as a function of prototype mismatch. We also show, qualitatively and quantitatively, that when there is a mismatch between the true and prototype behaviors, the diffeomorphism tends to overfit the data. In such cases, training multiple diffeomorphisms on random 50% data splits recovers highly varying invariant sets. The variance in the estimated invariant sets, together with the magnitude of the loss, can be used in order to detect prototype mismatch, and also allows for a measure of uncertainty. These results are now also part of Appendix F4 (Fig. 15).

---

### Official Review · Reviewer_Pr4v · 2025-10-31

**Soundness:** 2
**Presentation:** 3
**Contribution:** 3
**Rating:** 4
**Confidence:** 3

**Summary:**

This paper proposes a Smooth Prototypical Equivalence (SPE) framework, a model designed to predict the invariant sets of unknown dynamical systems by pushforwarding known, simple prototypical vector fields. The model learns diffeomorphisms, parameterized by normalizing flows, that smoothly map between vector field data sampled from unknown systems and their prototypical counterparts. If a system is smoothly (or orbit) conjugate to a specific prototypical vector field, then the pushforward of the observations should match that prototypical vector field. This principle is encoded in the equivalence loss, which measures the discrepancy between the two. The prototypical vector field with the lowest equivalence loss is then selected as the canonical form of the data, and the pullback of its invariant set serves as an approximation of the invariant set of the data. The proposed method is benchmarked on several synthetic examples as well as a more complex single-cell gene expression dataset.

**Strengths:**

Matching the pushforward vector field with certain prototypes via smooth conjugacy is novel and conceptually sound. The paper is well-organized and easy to follow.

**Weaknesses:**

However, it is not entirely clear whether the smooth/orbit equivalence loss can be reliably applied in more general settings. When the equivalence loss is exactly zero, the pullback of the prototypical invariant set indeed corresponds to the true invariant set of the data dynamics. Yet, when the loss is nonzero, it is not guaranteed that a smaller equivalence loss implies a closer approximation of the true invariant set.

If the underlying invariant sets are hyperbolic, the equivalence loss might still be meaningful. The persistence theorem ensures that hyperbolic invariant sets cannot be destroyed by small perturbations. Thus, if the equivalence loss can be regarded as a perturbation metric, one might argue that a smaller loss increases the likelihood that the pullbacked invariant set remains approximately invariant. However, since the persistence theorem requires $C^1$-closeness, this justification is not entirely rigorous in the current formulation based on C0-closeness. It may be possible for the authors to derive a $C^1$ bound from $C^0$-closeness, though this is uncertain.

Moreover, if the underlying invariant sets are non-hyperbolic, no such guarantee exists. Even a small perturbation (= a nonzero equivalence loss) can drastically alter the structure of the invariant sets. The benchmarked systems in the paper appear to be restricted to low-dimensional, hyperbolic cases (such as attracting limit cycles), where the theoretical assumptions implicitly hold. It remains unclear whether the proposed framework would perform reliably beyond these settings.

**Questions:**

The authors' method relies on the assumption that a smaller equivalence loss implies a closer correspondence between the true invariant set of the target system and the pullbacked one. However, this relationship is not theoretically justified in the paper. What theoretical guarantee does your equivalence loss provide regarding the recoverability or approximation quality of invariant sets? Specifically,

- Can the equivalence loss be interpreted as a meaningful bound or metric (e.g., in the persistence theorem sense) on the deviation between true and estimated invariant sets?
- If not, under what conditions (e.g., hyperbolicity, structural stability) can a smaller equivalence loss be expected to correspond to a more accurate recovery of the invariant structure?

---

> ### Author Response · Authors · 2025-11-20
>
> We thank the reviewer for their appreciation of our approach as being ‘novel and conceptually sound’, and the manuscript being ‘well-organized and easy to follow’, and for their constructive theoretically oriented comments and suggestions. We have edited and extended the revised manuscript accordingly. Below we address each point:
>
> ---
>
> > _“If the underlying invariant sets are hyperbolic, the equivalence loss might still be meaningful...”_
>
> We agree with the reviewer regarding the value for a more rigorous demonstration of our loss’s use as a metric between vector fields with perturbed invariant sets. Following this suggestion, we provide a new analysis (Appendix E1)  of the conditions under which our loss acts as a continuous signal for vector field distance, and we have shown how this distance decays as the true perturbation between fields goes to zero. We believe this justifies our approach as not just a binary signal of equivalence but a continuous metric on vector fields.
>
> ---
>
> > _“Moreover, if the underlying invariant sets are non-hyperbolic, no such guarantee exists. Even a small perturbation (= a nonzero equivalence loss) can drastically alter the structure of the invariant sets. The benchmarked systems in the paper appear to be restricted to low-dimensional, hyperbolic cases (such as attracting limit cycles), where the theoretical assumptions implicitly hold. It remains unclear whether the proposed framework would perform reliably beyond these settings.”_
>
> We thank the reviewer for this important, theoretically motivated comment. As mentioned, we were primarily concerned with invariant sets that are restricted to be both low-dimensional and hyperbolic. This is because in our domain of interest, biological dynamical systems (e.g. cell cycle, circadian dynamics, differentiation drifts exhibited by cellular populations in single-cell data), the invariant sets typically observed are indeed hyperbolic (e.g., stable limit cycles and fixed points, corresponding to different, important classes of biological processes). Moreover, our data-driven approach is concerned with sparse, noisy observations from the full phase space. As per your comment, the localization of non-structurally stable invariant sets cannot be guaranteed using our framework, or perhaps any other, under such significant perturbations introduced by both sparsity and the addition of noise.
>
> We have added a discussion of this important point to Section 3.1 and the Discussion of the revised manuscript, to make these working assumptions regarding the true underlying invariant set transparent in the text.
>
> ---
>
> > _“Can the equivalence loss be interpreted as a meaningful bound or metric (e.g., in the persistence theorem sense) on the deviation between true and estimated invariant sets?” and “If not, under what conditions (e.g., hyperbolicity, structural stability) can a smaller equivalence loss be expected to correspond to a more accurate recovery of the invariant structure?”_
>
> We expect our method to work when the dynamics underlying the observed data, and the prototype, are both structurally stable, as we now demonstrate mathematically in Appendix E2 in the revised manuscript. In short, we consider the case where the prototype is equivalent to the underlying vector field. In this scenario, we explicitly show that the prototype together with the learned diffeomorphism can be viewed as a perturbation of the true underlying vector field, and that our equivalence loss is directly related to the size of this perturbation. Therefore, in these regimes, a larger difference between the vector fields (in the $C^0$ sense) implies a larger equivalence loss.  When the equivalence loss is low, it can be further shown that the prototype-diffeomorphism pair is $C^1$ perturbation of the true underlying vector field. This is true whenever all elements of the partial derivatives up to order two of the diffeomorphism are bounded. We have formulated these results mathematically, and we have added them to Appendix E2.
>
> Beyond these theoretical considerations, however, we believe that we have empirically demonstrated that our approach is able to estimate the position of the invariant set, both in simulated systems (which we have supported quantitatively) and in real biological data (which we have supported with biological interpretation of the reconstructed invariant set within gene expression space). In particular, we showed that even when the phase space is noisy and only partially observed, our method is still able to estimate the invariant set with prototypes that are smoothly equivalent to the true system.

---

### Author Response · Authors · 2025-12-01

We  thank  the  reviewers  for  their  constructive comments and overall positive  evaluation of our manuscript, stating that “_matching the pushforward vector field with certain prototypes via smooth conjugacy is **novel and conceptually sound** [and] the paper is well-organized and easy to follow_” (Reviewer Pr4v), describing our work as “_**robust, outperforming existing techniques**, especially in realistic, data-scarce, and noisy scenarios_”,  which we used to “_successfully apply this method to high-dimensional biological data_”, scalable (“_The extraction of a biologically meaningful trajectory from 100D real-world cell cycle data is a **particularly impressive** result_”), and interpretable (“_**[offering] scientists deeper insights**_”)  (Reviewer 1nY5), and describing our contributions as “_novel_”, writing that “_the use of prototypes … provides **clear physical and qualitative meaning to the learned mappings**_”, which we used to demonstrate that “_[the] method can infer invariant structures and classify system behaviors directly from sparse vector-field data without knowing governing equations_”, and that the method’s **scalability and inference in real single-cell data** “_shows **practical value in biological science** in a data-driven approach_” (Reviewer AQH7).


We have addressed all comments and suggestions raised by the reviewers, in a point-by-point response to each reviewer, and have revised our manuscript and appendix accordingly. The main points of the revision are:

---

1. Following the suggestion of Reviewer Pr4v, we rigorized the theoretical underpinning of our work, showing that lower equivalence losses truly correspond with more faithful reconstructions of the invariant set (new Appendix E).

---

2. We addressed the shared comments of Reviewers 1nY5 and AQH7 regarding the ability of SPE to discern when the true system is not part of the chosen dictionary of prototypes by adding experiments and related discussion to the revised manuscript, demonstrating that SPE is able to distinguish dynamic behaviors within and outside the prototype dictionary (new Appendices F2 and F4).

---

3. Following suggestions by Reviewers 1nY5 and AQH7, we have added to the revised manuscript further ablation studies on the architecture of the invertible neural network, as well as motivation for the chosen network and guidelines for choices of hyperparameters (Appendix A8).

---

4. We have demonstrated the ability of SPE to match more complicated systems (Appendix F1), and have also expanded future directions for matching more complex behaviors in the Discussion section (following requests by Reviewer 1nY5).

---

5. Following a question raised by Reviewer AQH7, we extended the discussion on the choice of prototypes in Appendix F3.

---

We believe that these demonstrations and extensions further strengthen our work.

---

### Meta-Review · Area_Chair_hGNY · 2026-01-05

**Summary:**

All reviewers agreed that the proposed methodology of classification via prototype equivalence as an interesting and novel direction for the data-driven study of dynamical systems. Moreover, the data-efficiency and favourable scaling to higher dimensions are desirable properties. A common weakness identified is the need for an a priori determined dictionary of prototypes, which in general may be difficult to obtain in a principled manner. In addition, Reviewer Pr4v raised a concern of whether a small loss (eq 2) implies the closeness of invariant sets. This appears to rely crucially on hyperbolicity.

**Reviewer Concerns:**

The authors added a theoretical analysis which basically shows that by assuming hyperbolicity, minimising the proposed loss gives C1 closeness of vector fields, thus closeness of invariant sets. This partially addresses Reviewer Pr4v’s concern, but not completely. While it is true most interesting invariant sets are hyperbolic, very often one wants to study bifurcation behaviour where non-hyperbolic behaviour is present at the bifurcation points. In fact, it is precisely systems with bifurcations that one often wants to classify invariant behaviour as the bifurcation parameter varies. In this sense, the analysis in appendix E is not sufficient to ensure that the results can be reliable, because it lacks quantitative estimates of the distance of invariant sets vs the magnitude of the loss (it is a continuity-type result). I think this can be improved.

Reviewers also raised the issue of uniqueness of learned diffeomorphisms. I do not believe that the authors addressed this correctly in the mathematical sense. The authors mention that the neural network parameters are non-unique and noise may cause non-indentifiability. While correct, the more fundamental issue is that for two smoothly equivalent dynamical systems, in general there does not exist a unique map between them (centralisers of both source and target flows cause non-uniqueness).

Overall, the revision and rebuttal process improved the paper, which has good empirical results. The theoretical results can be strengthened, however.

**Reviewer Scores:**

The reviewers are likely to slightly raise their score in view of the clarifications, but the paper remains borderline.

---

### Decision · Program_Chairs · 2026-01-26

Reject